# Operator Learning with Domain Decomposition for Geometry Generalization in PDE Solving

**Jianing Huang**[✉],[*] **Kaixuan Zhang, Youjia Wu, Ze Cheng**
Corporate Research, Bosch (China) Investment Co., Ltd.
`{jianing.huang, kaixuan.zhang, youjia.wu, ze.cheng}@cn.bosch.com`

## Abstract

Neural operators have become increasingly popular in solving *partial differential equations* (PDEs) due to their superior capability to capture intricate mappings between function spaces over complex domains. However, the data-hungry nature of operator learning inevitably poses a bottleneck for their widespread applications. At the core of the challenge lies the absence of transferability of neural operators to new geometries. To tackle this issue, we propose operator learning with domain decomposition, a local-to-global framework to solve PDEs on arbitrary geometries. Under this framework, we devise an iterative scheme *Schwarz Neural Inference* (SNI). This scheme allows for partitioning of the problem domain into smaller subdomains, on which local problems can be solved with neural operators, and stitching local solutions to construct a global solution. Additionally, we provide a theoretical analysis of the convergence rate and error bound. We conduct extensive experiments on several representative linear and nonlinear PDEs with diverse boundary conditions and achieve remarkable geometry generalization compared to alternative methods. These analysis and experiments demonstrate the proposed framework's potential in addressing challenges related to geometry generalization and data efficiency. The code is publicly available at this repository: `https://github.com/questionstorer/sni`.

## 1 Introduction

*Partial differential equation* (PDE) solving is of paramount importance in comprehending natural phenomena, optimizing engineering systems, and enabling multidisciplinary applications (Evans, 2022). The computational cost associated with traditional PDE solvers (Liu and Quek, 2013; Lu et al., 2019) has prompted the exploration of learning-based methods as potential alternatives to overcome these limitations. Neural operators (Li et al., 2020b; 2023; 2024; Liu et al., 2023; Hao et al., 2023), as an extension of traditional neural networks, aim to learn mappings between the functional dependencies of PDEs and their corresponding solution spaces. They offer highly accurate approximations to classical numerical PDE solvers while significantly improving computational efficiency. Despite its success, operator learning, as a data-driven approach, encounters the inherent 'chicken-and-egg' problem, revealing an interdependence between operator learning and the availability of data. This dilemma arises from the challenge of simultaneously addressing the inefficiency of classical solvers and acquiring an ample amount of data for neural operator training.

Existing works in alleviating the above challenges explore symmetries of PDEs. Lie point symmetry data augmentation (LPSDA) (Brandstetter et al., 2022) generates potentially infinitely many new solutions of PDE from existing solution by exploiting symmetries of differential operator defining the PDE. Subsequent work (Mialon et al., 2023) applies LPSDA for self-supervised learning. However, LPSDA only partially alleviate the problem in data efficiency and the problem of how to quickly generalize to new geometry is untouched. While existing neural operators have shown capabilities in handling diverse geometries through approaches such as geometry parametrization (Li et al., 2023) or coordinate representation (Hao et al., 2023), they lack the ability to generalize to entirely novel geometries that differ significantly from those present in the training data distribution. The inability

---

[*]Corresponding author: jianing.huang@cn.bosch.com

to quickly adapt neural operators to unseen geometries without further generating new data hinders the applicability of neural operator learning to real-world problems in industry.

To tackle this challenge, a natural idea is to break down a domain into some basic shapes where neural operator can generalize well. *Domain decomposition methods* (DDMs) (Toselli and Widlund, 2004; Mathew, 2008) provide the suitable tool for this purpose. Related efforts such as Mao et al. (2024) have combined operator learning with DDMs on uniform grids to accelerate classical methods. In contrast, our work aims to extend this paradigm to arbitrary geometries through a local-to-global framework. This framework consists of three parts: (1) Training data generation: creation of random basic shapes and imposition of appropriate boundary conditions on these shapes. This generated data serves as the training set for the neural operator in our framework. (2) Local operator learning: neural operator training to learn solutions on basic shapes. Data augmentation based on symmetries of PDEs is utilized to enable the neural operator to capture the intricate details and variations within these shapes. (3) Schwarz neural inference (SNI): a three-step algorithm for inference. Firstly, the computational domain is partitioned into smaller subdomains. Then, the learned operator is applied within each subdomain to obtain the local solution. Finally, an iterative process of stitching and updating the global solution is performed using additive Schwarz methods.

**Our Contributions.** We summarize our contributions below:

- We introduce a local-to-global framework that integrates operator learning with domain decomposition methods as an attempt in tackling the geometry generalization challenge in operator learning.

- We design a novel data generation scheme that leverages random shape generation and symmetries of PDEs to train local neural operators for solving PDEs on basic shapes.

- We propose an iterative inference algorithm, SNI, built upon a trained local neural operator to obtain solutions on arbitrary geometries. We theoretically analyze the convergence and the error bound of the algorithm for a wide range of elliptic PDEs. Through comprehensive experiments, we empirically validate the effectiveness of our framework on generalizing to new geometries for both linear and nonlinear PDEs.

## 2 PROBLEM FORMULATION AND PRELIMINARIES

In this section, we provide an introduction to the problem formulation and essential background on domain decomposition methods, which will be utilized throughout the entirety of the paper.

### 2.1 PROBLEM FORMULATION

Our primary focus is on stationary problems of PDEs defined in the following form:

$$\begin{aligned}
\mathcal{L}(u) &= f && \text{in } \Omega \\
u &= u_D && \text{on } \Gamma_D \\
\frac{\partial u}{\partial n} &= g && \text{on } \Gamma_N
\end{aligned} \tag{1}$$

where $\mathcal{L}$ is a partial differential operator and $\Gamma_D \cup \Gamma_N = \partial\Omega$ denotes Dirichlet and Neumann boundary, respectively. We assume all the domains $\Omega$ are bounded orientable manifolds embedded in some ambient Euclidean space $\mathbb{R}^n$ (Li et al., 2023). Later we will extend our method to handle time-dependent equations.

We consider situations where geometry of domain $\Omega_{\text{inf}}$ at inference time is decoupled from that of $\Omega_{\text{train}}$ in training time, i.e., $\Omega_{\text{inf}}$ does not have to fall in or resemble training geometries and can be of arbitrary shapes. For implementation we will mainly focus on $\Omega \subseteq \mathbb{R}^2$.

### 2.2 DOMAIN DECOMPOSITION METHODS

Domain decomposition methods (DDMs) solve Eq. 1 by decomposing domain into subdomains and iteratively solve a coupled system of equations on each subdomain. An *overlapping decomposition* of

$\Omega$ is a collection of open subregions $\{\Omega_k\}_{k=1}^K$, $\Omega_k \subseteq \Omega$ for $k = 1, \ldots, K$ such that $\bigcup_{k=1}^K \Omega_k = \Omega$. We denote $V$ and $\{V_k\}_{k=1}^K$ to be finite element space associated with domain $\Omega$ and $\{\Omega_k\}_{k=1}^K$. We can define *restriction operators* $\{R_k : V \to V_k\}_{k=1}^K$ restricting functions on $\Omega$ to $\{\Omega_k\}_{k=1}^K$ and *extension operators* $\{R_k^\mathsf{T} : V_k \to V\}_{k=1}^K$ extending functions on $\{\Omega_k\}_{k=1}^K$ to $\Omega$ by zero.

In the subsequent discussion, we revisit the idea of *additive Schwarz method* (ASM) in DDMs for overlapping decomposition. The *additive Schwarz-Richardson iteration* (Mathew, 2008) has the following form:

$$u^{n+1} = u^n + \tau \sum_{k=1}^K \left[ R_k^\mathsf{T} w_k^{n+1} - R_k^\mathsf{T} R_k u^n \right] \tag{2}$$

where $0 < \tau < \frac{1}{K}$ is a hyperparameter controlling the convergence rate, and $w_k^{n+1}$ is the solution of the following equation:

$$
\begin{aligned}
\mathcal{L}(w_k^{n+1}) &= 0 && \text{in } \Omega_k \\
w_k^{n+1} &= u_D && \text{on } \partial\Omega_k \cap \Gamma_D \\
\frac{\partial w_k^{n+1}}{\partial n} &= g && \text{on } \partial\Omega_k \cap \Gamma_N \\
w_k^{n+1} &= u^n && \text{on } \partial\Omega_k \cap \Omega
\end{aligned}
\tag{3}
$$

We denote the local operator $\mathcal{S}_k : (u^n, u_D, g) \mapsto w_k^{n+1}$. Note that the first two boundary conditions in Eq. 3 is the boundary condition on the global boundary part of $\partial\Omega_k$ and is not updated during iteration. The last boundary condition is along the artificial boundary created by decomposition and the value is updated through iteration. Hence $\{\mathcal{S}_k\}_{k=1}^K$ can be considered as a single-input operator when the global boundary condition and decomposition are determined. This iterative process can be shown to converge for FEMs under mild assumption on properties of equation and decomposition. Please refer to Appendix A for more details.

## 3 OPERATOR LEARNING WITH DOMAIN DECOMPOSITION

In order to solve PDE on arbitrary geometry with neural operator, a natural idea is to decompose domain into a prescribed family of building blocks (basic shapes) since it is not feasible to explicitly consider arbitrary shapes during training stage. For that purpose, we propose to train a neural operator to solve local problems on basic shapes and stitch local solutions together to get a global solution. An illustration of the proposed framework is presented in Figure 1. A detailed implementation will be discussed in the following subsections.

### 3.1 TRAINING DATA GENERATION

Data generation serves the purpose of operator learning, which fundamentally aims to approximate the local solution operator $\mathcal{G} : \mathcal{P} \times \mathcal{H} \to \mathcal{U}$. Here, $\mathcal{P}$ denotes the space of basic shapes, $\mathcal{H}$ represents boundary conditions and other input functions, $\mathcal{U}$ represents the solution space. Next we will delve into a comprehensive examination of how $\mathcal{P}$ and $\mathcal{H}$ are determined separately.

**Choice of basic shapes.** The selection of basic shapes cannot be arbitrary due to the requirement of ensuring the neural operator's capability in solving local problems across a wide range of shapes. Moreover, three necessary criteria should be set forth for basic shape generation: (1) sampling feasibility: it should be tractable to sample from and solve boundary value problems on basic shapes in $\mathcal{P}$. (2) complete coverage: basic shapes in $\mathcal{P}$ should be flexible to cover any shape of domain.

For implementation, we focus on $\Omega \subseteq \mathbb{R}^2$. We propose to use the space $\mathcal{P}_s(n)$ of simple polygons with at most $n$ vertices (i.e. planar polygon without self-intersection and holes) uniformly bounded by a compact region in $\mathbb{R}^2$. Simple polygons are Lipschitz domains with straightforward sampling method (Auery and Heldz, 2019) and flexible enough to constitute any discretized planar domain (Preparata and Shamos, 2012). We note, however, that this is not the only choice of these basic shapes. We could equally use convex polygons, star-shaped polygons, etc. as long as the two aforementioned criteria are satisfied.

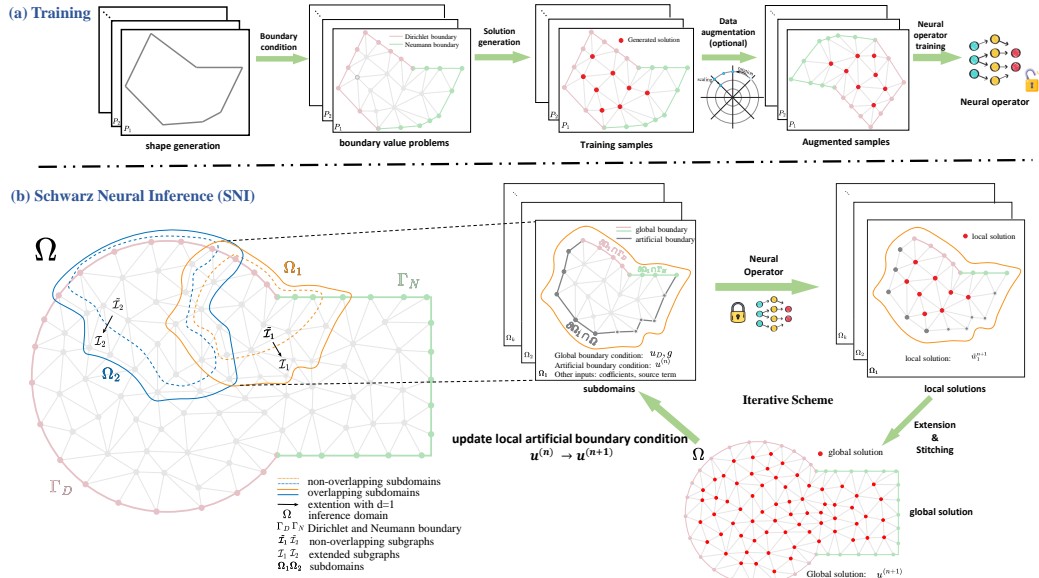

Figure 1: An illustration of Operator Learning with Domain Decomposition Framework. (a) During training stage, the goal is to ensure that the neural operator can effectively model the local solution operator on various building blocks of shapes. These building blocks are selected and generated based on specific criteria, allowing for a more efficient and targeted learning process. Proper boundary conditions are then imposed to generate local solutions which serve as training data for neural operator. (b) During inference, for an arbitrary given domain, an automated decomposition algorithm is employed to decompose the domain into subdomains. By leveraging the trained local operator and Schwarz Neural Inference (SNI), global solution can be obtained by stitching local solutions on subdomains.

**Imposing boundary conditions.** The imposition of boundary conditions presents two complications: (1) Types of boundary conditions. Neumann boundary conditions in Eq. 1 will inevitably result in mixed boundary conditions in local subdomains. To generate solutions with mixed boundary conditions, we randomly divide the boundary of a basic shape into two connected components, representing the Dirichlet and Neumann boundaries, respectively. During inference, we have to carefully set hyperparameters for decomposition to make sure boundary of subdomains have at most two connected components for Dirichlet and Neumann boundaries. (2) Functional range of boundary conditions. In general, the inference process for subdomains will encounter arbitrary ranges in boundary conditions. However, it is practically infeasible to train the neural operator to handle unbounded boundary values. Instead, we generate random functions with values normalized within a bounded range for both boundary conditions and other input functions such as coefficient fields and source terms. We will handle this complication with symmetries of PDEs during inference.

## 3.2 LOCAL OPERATOR LEARNING

We now train a neural operator $\mathcal{G}^\dagger$ to approximate the mapping $\mathcal{G}$. Our focus is not on design of neural operators, but on ensuring that the neural operator can solve local problems accurately.

**Choice of neural operator architecture.** Our framework is orthogonal to the choice of neural operator architecture as long as the architecture can accommodate flexible input/output formats and possesses sufficient expressive power to solve local problem with randomly varying domain and input functions. For implementation, we adopt GNOT (Hao et al., 2023) which is a highly flexible transformer-based neural operator. We note that, however, training neural operator on highly varying geometries presents challenges to both design of architectures and training schemes.

**Data augmentation.** To enhance the generalization capabilities of the neural operator, Lie point symmetry data augmentation (LPSDA) (Brandstetter et al., 2022) can be naturally applied to local

---

**Algorithm 1** Schwarz Neural Inference

---

**Input:** Domain $\Omega$; Global boundary data $B$ (Dirichlet/Neumann); Other input functions $H$ (e.g., coefficient fields, source terms, PDE parameters); Number of subdomains $K$; Depth of extension $d$; Local operator $\mathcal{G}^\dagger$; Step size $\tau$; Convergence criterion $C$;

**Output:** Global Solution $u$;

1: Apply METIS and extension to get overlapping decomposition $\{\Omega_k\}_{k=1}^K$, obtain restriction operators $\{R_k\}_{k=1}^K$ and extension operators $\{R_k^\intercal\}_{k=1}^K$;
2: Initialize the global solution $u^0$;
3: **while** convergence criterion $C$ not satisfied **do**
4:     Form local inputs $\{(\Omega_k, B_k^n, H_k)\}_{k=1}^K$ from $\Omega$, $B$, $H$ and the last-step solution $u^n$;
5:     Obtain preprocessing transforms $\{T_k\}_{k=1}^K$ and postprocessing transforms $\{\tilde{T}_k\}_{k=1}^K$;
6:     Local inference on each subdomain: $\hat{w}_k^{n+1} = \tilde{T}_k\big(\mathcal{G}^\dagger\big(T_k(\Omega_k, B_k^n, H_k)\big)\big)$;
7:     Global update (additive Schwarz form): $u^{n+1} = u^n + \tau \sum_{k=1}^K R_k^\intercal\big(\hat{w}_k^{n+1} - R_k u^n\big)$;
8:     $n = n + 1$;
9: **end while**
10: **return** $u^n$;

---

solutions during training. Examples of such transformations are rotation and scaling. It is crucial to appropriately extend these transformations to boundary conditions and other input functions, taking into account the symmetries inherent in the PDEs. Please refer to Appendix E for a detailed discussion.

### 3.3 SCHWARZ NEURAL INFERENCE

Inspired by additive Schwarz method, we introduce a similar iterative algorithm called Schwarz Neural Inference (SNI), which is outlined in Algorithm 1. In the subsequent discussion, we will explore several important considerations.

At each iteration $n$, SNI assembles, for every subdomain $\Omega_k$, a local boundary input $B_k^n$ by combining the prescribed global boundary data on $\partial\Omega_k \cap \partial\Omega$ and the interface Dirichlet data from the previous iterate on $\partial\Omega_k \cap \Omega$. For PDEs with additional input functions or parameters (e.g., coefficient fields and source terms), we denote by $H_k$ their restriction to $\Omega_k$. In Algorithm 1, $\hat{w}_k^{n+1}$ denotes the post-processed local solution on $\Omega_k$.

**Decomposition into overlapping subdomains.** In general, there is no canonical way to decompose an arbitrary domain into prescribed shapes. Here we adopt a common practice in the DDM literature (Mathew, 2008). We assume there exists a pre-defined triangulation $\mathcal{T}_h(\Omega)$ of the domain $\Omega$, and a graph can be constructed to represent the connectivity of this triangulation. A graph partition algorithm such as METIS (Karypis and Kumar, 1997) is then employed to partition this graph into $K$ non-overlapping connected subgraphs with index sets $\tilde{\mathcal{I}}_1, \ldots, \tilde{\mathcal{I}}_K$. To achieve an overlapping decomposition, each subgraph is then extended iteratively by including neighboring vertices for $d$ iterations. This process generates index sets $\mathcal{I}_1, \ldots, \mathcal{I}_K$ that, together with the original mesh, form an overlapping decomposition denoted as $\{\Omega_k\}_{k=1}^K$. An intuitive illustration of this process is depicted in Figure 1.

For implementation, partition number $K$ and extension depth $d$ are hyperparameters that should be carefully set to ensure that the resulting subdomains resemble shapes in $\mathcal{P}$.

**Normalization.** During inference on an arbitrary decomposed subdomain, the range of geometry and boundary conditions may differ from that of the generated training data. We thus leverage the symmetry properties of PDEs to handle this mismatch. More specifically, we can apply problem-dependent transformations $T_k$ (e.g., spatial translation/scaling and value shift/scaling) to map a local input $(\Omega_k, B_k^n, H_k)$ that lies outside the training range back into the training range. When the PDE symmetry acts on additional inputs (e.g., coefficient fields, source terms, parameters), the transformation is extended to $H_k$ accordingly. After neural operator inference, we map the predicted local solution back via an inverse transformation $\tilde{T}_k$. We implement these as preprocessing and postprocessing steps in the inference pipeline (see Appendix E).

**Time Complexity.** Suppose the single inference time of local operator and the number of iterations are denoted as $b$ and $N$. Let $v, e, K$ denote the number of vertices, edges and subdomains respectively. Our Algorithm 1 consists of two main parts: mesh partition using the METIS algorithm, the time complexity of which is approximately $O(v + e + K \log K)$ (Karypis and Kumar, 1997); and an additive-Schwarz-style iterative scheme with a time complexity roughly $O(bKN)$. Therefore, the overall time complexity of our algorithm can be approximated as $O(v + e + K \log K + bKN)$[1]. In practice, $K$ may not be independent of $v$ and $e$. More vertices can sometimes lead to more partitions required depending on the property of mesh. While providing an exact time complexity analysis for FEM can be challenging due to the complexity and variability of different problem setups, it is worth noting that FEM is generally considered to be computationally demanding.

### 3.4 THEORETICAL RESULTS

Here we provide a theoretical analysis of our proposed algorithm by stating the following result:

**Theorem 1.** *Assume the operator $\mathcal{L}$ in Eq. 1 is self-adjoint and coercive elliptic (Mathew, 2008). Let $u^n$ be the iterates of the classical additive Schwarz-Richardson iteration in Eq. 2 and let $\tilde{u}^n$ be the iterates of SNI in Algorithm 1, initialized with the same $u^0 = \tilde{u}^0$.*

*Let $\mathcal{S}_k$ denote the exact local solution operator on $\Omega_k$, and define the learned (pre/post-processed) local solver $\tilde{\mathcal{S}}_k \triangleq \tilde{T}_k \circ \mathcal{G}^\dagger \circ T_k$. Assume:*

- *(Classical contractivity) the classical iteration mapping is contractive in a norm $\|\cdot\|$ with rate $0 < \rho < 1$;*

- *(Uniform local error) $\|\tilde{\mathcal{S}}_k(\cdot) - \mathcal{S}_k(\cdot)\| \leq c$ for all $k$;*

- *(Overlap factor) each degree of freedom is covered by at most $t$ subdomains.*

*Then we have:*

- ***Convergence:** SNI converges to a fixed point $\tilde{u}^*$.*

- ***Error bound:** for all $n$, $\|\tilde{u}^n - u^n\| \leq c'$, and $\|\tilde{u}^* - u^*\| \leq c'$, where one may take*

$$c' = \frac{\tau t}{1 - \rho} c.$$

The theorem suggests that if our learned local operator maintains a uniform error bound, the algorithm converges and exhibits a minimal approximation error. See Appendix C for a proof. This result relies on the assumption on operator $\mathcal{L}$. In general, such convergence is not guaranteed and we empirically validate the effectiveness of our framework for nonlinear differential equation through experiment.

## 4 EXPERIMENTS

In this section, we perform comprehensive experiments to showcase the effectiveness of our method on various challenging datasets.

### 4.1 EXPERIMENTAL SETUP

**Datasets.** To demonstrate the scalability and superiority of our method, we construct several datasets on multiple PDEs. We also extend our framework to a time-dependent problem, heat conduction. To aggregate training sets, we generate random simple polygons bounded by the unit square $[-0.5, 0.5]^2 \subset \mathbb{R}^2$. Boundary/initial conditions and coefficient functions are piecewise linear functions determined by random values within $[0, 1]$. For each of the following problems, we test on datasets based on three different domains A, B and C shown in Figure 2. Details of these datasets are given in Appendix H.1.

---

[1]With neural operators implementing linear transformers, e.g., GNOT applied in this work, $b = O(\frac{v}{K})$.

| Equation | Domain | GNOT(%) | SNI(%) |
|----------|--------|---------|--------|
| Laplace2d-Dirichlet | A | 22±2 | 2.2±0.6 |
| | B | 22±2 | 2.1±0.4 |
| | C | 28±3 | 2.1±0.9 |
| Laplace2d-Mixed | A | 10.7±0.8 | 6±4 |
| | B | 10.7±0.8 | 7±1 |
| | C | 38±6 | 6±1 |
| Darcy2d | A | 16±1 | 8±2 |
| | B | 63±3 | 8±2 |
| | C | 167±8 | 5.4±0.6 |
| Heat2d | A | 11.5±0.6 | 5.3±0.2 |
| | B | 30±10 | 11±2 |
| | C | 20±10 | 5.8±0.3 |
| NonlinearLaplace2d | A | 22±2 | 2.0±0.4 |
| | B | 26±2 | 2.2±0.4 |
| | C | 28±2 | 2.2±0.5 |

Table 1: Main results. The $l_2$ relative errors along with standard deviation over different random boundary/initial conditions on three domains are reported.

- **Laplace2d-Dirichlet**: Laplace equation in 2d with pure Dirichlet boundary condition on various shapes.

- **Laplace2d-Mixed**: Laplace equation in 2d with mixed Dirichlet and Neumann boundary condition on various shapes.

- **Darcy2d**: Darcy flow in 2d with coefficient field $a(x)$, source term $f(x)$ and pure Dirichlet boundary condition on various shapes.

- **Heat2d**: Time-dependent heat equation in 2d with a coefficient $\alpha$ for thermal diffusivity, initial condition and time-varying pure Dirichlet boundary condition on various shapes.

- **NonlinearLaplace2d**: A nonlinear Laplace equation in 2d with pure Dirichlet boundary condition on various shapes.

**Baseline.** Our baseline is a direct inference of the trained neural operator on domains shifted and scaled to $[-0.5, 0.5]^2$ with boundary/initial conditions and coefficient functions adjusted accordingly.

**Evaluation Protocol.** The evaluation metric we utilize is the mean $l_2$ relative error. See Appendix H.2 for details.

## 4.2 Main Results and Analysis

The main results for all datasets are shown in Table 1. More details and hyperparameters are summarized in Appendix H.2 due to limited space. Based on these results, we have the following observations.

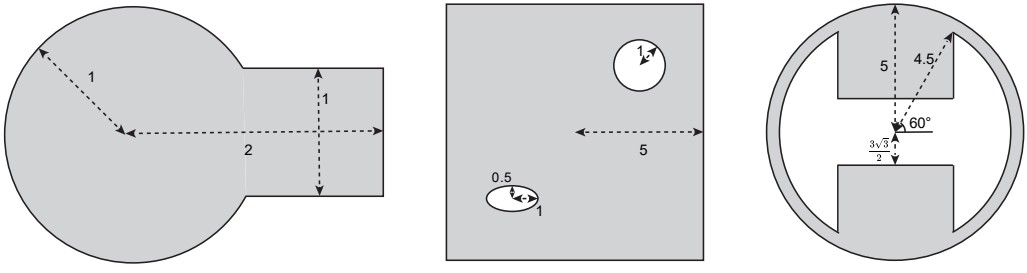

Figure 2: Illustration of experiment domain A, B, C from left to right respectively.

**Stationary Problems.** First, we find that our method performs significantly better on all stationary problems compared with baseline. On all domains, we reduce prediction error by 34.8%-96.8%. The excellent performance shows the effectiveness of our framework in dealing with arbitrary geometries unseen during training. In particular, our framework usually leads by a larger margin on more complicated domain, due to the fact that simple polygons used in the training data fail to adequately resemble the complex testing domains. Solutions on multiply connected domains usually exhibit characteristics that are not present on simple domains.

Second, we find that the the performance of our method is consistent across various geometries during inference. On all types of PDEs in our datasets, the difference in prediction error over various geometries is within 3.25%, showing the ability to solve PDE with consistent accuracy on various geometries with a single trained neural operator. This also provides evidence for our theoretical result in Theorem 1 where we show that the SNI ensure the convergence to an approximation of the ground-truth solution with error bound determined by the generalization error of the neural operator.

Third, we find that complexity of the PDE together with types of boundary condition affect the generalizability of the neural operator in solving local problems and thus also the accuracy of our method. For simple problem such as Laplace2d-Dirichlet, our method achieve a 59.8% lower error compared to other problems. For Laplace2d-Mixed, neural operator struggles to capture subtlety in presence of both Dirichlet and Neumann boundaries. The complexity of Darcy2d lies in the need to capture changes in coefficient and source term in addition to geometry and boundary condition. We argue that having a strong neural operator that can generalize well on all basic shapes and boundary conditions is necessary for our framework to work with reasonable accuracy.

**Time-dependent Problems.** There is a natural way to extend our framework to time-dependent problems (Li and Cai, 2015) where a space-time decomposition is constructed by taking the product of a spatial decomposition and a temporal decomposition. We train a neural operator that can predict heat conduction on multiple time steps and the same SNI is applied during inference on this 3d problem. Our framework works well on this problem and reduce prediction error by 54.1%-74.2%. This demonstrates the potential of our framework to handle time-dependent problems. We refer to Appendix D for detailed implementation.

**Data Efficiency.** The exploration results on data efficiency of SNI are shown in Figure 3, implying the following observations: (1) At all abundances of data, the $l_2$ relative errors of SNI are significantly lower than those of GNOT direct inferences; (2) Errors of SNI are comparable to or even lower than validation errors at large data volumes. (3) SNI requires much smaller datasets to achieve comparable results to GNOT direct inference. Overall, these results demonstrate that SNI has substantial advantages in terms of data efficiency. Our proposed framework possesses remarkable ability to extract more insights from limited data and scale more effectively as data volumes increase. More supplementary results are provided in Appendix H.4.

## 4.3 ABLATION EXPERIMENTS

**Hyperparameter Exploration.** The number of partitions ($K$), the depth of extension ($d$) and step size ($\tau$) are the key main hyperparameters that can affect the performance of SNI. Based on the

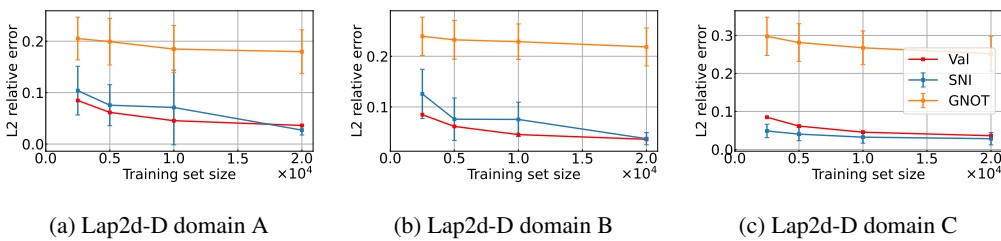

|  (a) Lap2d-D domain A | (b) Lap2d-D domain B | (c) Lap2d-D domain C |

Figure 3: Comparison between the $l_2$ relative errors from SNI (blue), GNOT direct inference (orange) on Laplace2d-Dirichlet upon three domains (A, B and C) with different numbers of training samples. The results of SNI and GNOT direct inference are presented based on 100 inferences with different boundary conditions. The best validation errors during training (red) are also provided as a reference.

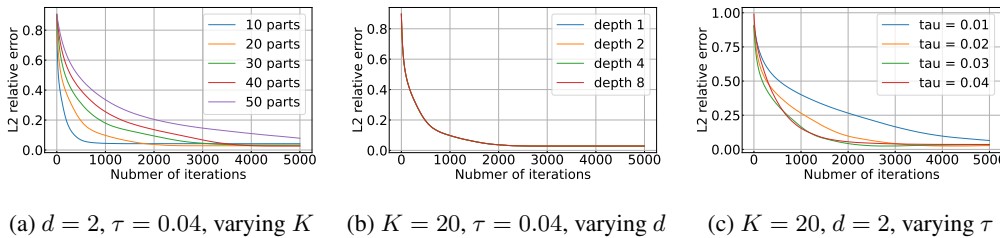

(a) $d = 2$, $\tau = 0.04$, varying $K$     (b) $K = 20$, $\tau = 0.04$, varying $d$     (c) $K = 20$, $d = 2$, varying $\tau$

Figure 4: Comparison between convergence rate of SNI on Laplace2d-Dirichlet domain A.

results presented in Figure 4, the factors analyzed have no significant impact on the accuracy of our algorithm, but they do influence the convergence rate. Specifically, increasing the number of partitions leads to a smaller $l_2$ relative error but slower convergence. Once the partition number surpasses 20, the algorithm's final performances become comparable. Regarding the depth of extension, it does not affect the performance on the tested domains. The convergence curves for depth of extension 1, 2, 4, and 8 are nearly identical. When it comes to $\tau$, a larger value results in faster convergence. However, it is important to note that there exists a maximum limit $1/K$ beyond which $\tau$ cannot be set.

**Data Augmentation Exploration.** To explore the effects of data augmentation, we compare the performances of models trained with different degrees of data augmentation for Laplace2d-Dirichlet demonstrated in Table 2. For models trained without data augmentation, the variation of performances on different domains is large, ranging from 2.8% to 4.4%. Specifically, it reports a $4.4 \pm 1.6\%$ $l_2$ relative error on domain A, while this error can be reduced to $1.9 \pm 0.4\%$ with a rotation+[0.8, 1] scaling augmentation. While rotation can generally be beneficial, the effectiveness of scaling can sometimes be limited or even detrimental. Hence, it is important to apply data augmentation with caution and consider its suitability for different types of PDEs.

**Choice of neural operator architecture.** To explore the choice of neural operator architecture in our framework, we train a Geo-FNO (Li et al., 2023) on Laplace2d-Dirichlet and apply SNI for inference on domains A, B and C to get $l_2$ relative error of $9 \pm 3\%$, $13 \pm 1\%$ and $13 \pm 3\%$. This result is comparable to that achieved by SNI with GNOT and demonstrates that our proposed framework works with various choices of neural operator architecture. However, an error gap does exists between SNI with GNOT and Geo-FNO due to variations in their generalizability. This is also reflected in their respective best validation errors, as detailed in Appendix H.4. Supplementary results on Darcy2d are also provided there.

## 5 RELATED WORK

**Operator Learning.** The idea of operator learning is first introduced in Lu et al. (2019). This work proposes a notable architecture called DeepONet, which employs a branch network for processing input functions and a trunk network for handling query points. Adopting the trunk-branch architecture and utilizing the attention mechanism, Hao et al. (2023) develops GNOT to handle irregular mesh, multiple input functions, and different input data types. The high accuracy and versatility makes GNOT the benchmark in our work. In the other direction, Fourier neural operator (FNO) (Li et al., 2020b) leverages the Fast Fourier Transform (FFT) to learn operators in the spectral domain, and

| | | Validation(%) | Domain A(%) | Domain B(%) | Domain C(%) |
|---|---|---|---|---|---|
| No Data Aug | | 3.79 | 4±2 | 3.0±0.6 | 3±1 |
| Rotation Only | | 2.50 | 2.2±0.6 | **2.1±0.4** | **2.1±0.9** |
| Rotation + Scale | [0.2, 1] | 5.31 | 4±1 | 3.4±0.5 | 3.4±0.4 |
| | [0.5, 1] | 3.62 | 2.7±0.5 | 3.7±0.6 | 3.2±0.6 |
| | [0.8, 1] | 2.86 | **1.8±0.4** | 3.3±0.7 | 2.8±0.8 |

Table 2: Comparison between models trained with different data augmentations for Laplace2d-Dirichlet.

achieves a favorable trade-off between cost and accuracy. Variants of FNO are proposed to reduce computational cost (FFNO in Tran et al. (2021)), handle irregular mesh (Geo-FNO in Li et al. (2023)), and improve expressivity (UFNO in Wen et al. (2022)).

**Methods to Deal with Complex Geometry.** Several approaches have been proposed to tackle the challenge of complex geometry and save the process efforts in operator learning. One encoder-process-decoder framework called CORAL (Serrano et al., 2023) is able to encode a complex geometry into a lower dimensional representation to save the computational efforts and solve different types of problems. In (Wu et al., 2024), one mechanism called physics attention is proposed to aggregate complex input geometry and functions into several physics-aware tokens to reduce the number of tokens to deal with. AROMA (Serrano et al., 2024) introduces a diffusion refiner in latent space to solve temporal problems with complex geometries.

**Domain Decomposition Methods Applied in Deep Learning.** In general, the integration of deep learning and DDMs can be categorized into two groups (Heinlein et al., 2021; Klawonn et al., 2024). The first category involves using deep learning techniques to improve the convergence properties or computational efficiency of DDMs. For instance, Mao et al. (2024) proposes to combine operator learning with DDMs on uniform grids in order to accelerate traditional DDMs. Several methods (Heinlein et al., 2020; 2019) have also been proposed to reduce the computational cost in adaptive FETI-DP solvers by incorporating deep neural networks while ensuring the robustness and convergence behavior. The second category is centered around the substitution of subdomain solvers in DDMs with neural networks. There have been multiple endeavors to employ PINNs (XPINNs (Jagtap and Karniadakis, 2020), parallel inference with cPINNs and XPINNs (Shukla et al., 2021)) or Deep Ritz methods as alternatives to subdomain solvers or discretization techniques in traditional DDMs (Li et al., 2020a; 2019; Jiao et al., 2021). These approaches leverage the universal approximation capabilities of neural networks to represent solutions of PDEs, subject to specific assumptions regarding the activation function and other factors.

**Data Augmentation Techniques in Operator Learning.** Different types of data augmentations are proposed to improve the generalization capabilities in operator learning. A Lie point symmetry framework is introduced in Brandstetter et al. (2022), which quantitatively derives a comprehensive set of data transformations, to reduce the sample complexity. Motivated by this approach, Mialon et al. (2023) learn general-purpose representations of PDEs from heterogeneous data by implementing joint embedding methods for self-supervised learning. An alternative research approach (Fanaskov et al., 2023) introduces a computationally efficient augmentation strategy that relies on general covariance and straightforward random coordinate transformations. In general, applying data augmentation techniques for PDE operator learning can be challenging due to the unique nature of PDE theory.

## 6  Conclusion and Future Works

We presented a local-to-global framework based on DDMs to address the geometry generalization and data efficiency issue in operator learning. Our framework includes a novel data generation scheme and an iterative inference algorithm SNI. Additionally, we provided a theoretical analysis of the convergence and error bound of the algorithm. We conducted extensive experiments to demonstrate the effectiveness of our framework and validate our theoretical result. For future works, the rich literature of DDMs when combined with operator learning provides many potential directions to handle higher-dimensional problems, non-overlapping decomposition and more challenging types of equations.

## REPRODUCIBILITY STATEMENT

Detailed descriptions of the experimental setup, task definitions, and evaluation metrics are provided in section 4 and Appendix H. Source code is attached in the submission.

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

## A    BACKGROUND ON DOMAIN DECOMPOSITION

Domain decomposition is a widely used technique in computational science and engineering that enables the efficient solution of large-scale problems by dividing the computational domain into smaller subdomains. This approach is particularly beneficial when dealing with complex problems that cannot be solved using a single computational resource. The main idea behind domain decomposition is to break down a large computational domain into smaller, more manageable subdomains. These subdomains can be arranged in a variety of ways, such as overlapping or non-overlapping, depending on the specific problem and the desired computational approach.

In this work, we decompose our domain into subdomains and adopt the hybrid formulation of Eq. 1 following Mathew (2008, Section 1.1). A *decomposition* of $\Omega$ is a collection of open subregions $\{\Omega_k\}_{k=1}^K$, $\Omega_k \subseteq \Omega$ for $k = 1, \ldots, K$ such that $\bigcup_{k=1}^K \overline{\Omega_k} = \overline{\Omega}$. This decomposition is referred to as *non-overlapping* if in addition, $\Omega_i \cap \Omega_j = \varnothing$ for any $i \neq j$. Alternatively, an *overlapping* decomposition is one satisfying $\bigcup_{k=1}^K \Omega_k = \Omega$. Typically, a non-overlapping decomposition is one where subdomains do not intersect with each other in the interior while an overlapping decomposition constructed in practice has overlapping neighboring subdomains.

Given a decomposition of $\Omega$, a *hybrid formulation* of Eq. 1 is a coupled system of local PDEs on subdomains $\Omega_k$ equivalent to Eq. 1 satisfying two requirements. First, the restriction $u_k(x)$ of the solution $u(x)$ of Eq. 1 to each domain $\Omega_k$ must solve the local PDE, thus ensures that the hybrid formulation is *consistent* with the original problem in Eq. 1. Second, the hybrid formulation must be *well posed* as a coupled system of PDEs in the sense of Evans (2022), i.e. its solution must exist, be unique and depend continuously on given input function and boundary/initial conditions. Intuitively, a hybrid formulation consists of a *local problem* posed on each subdomain and *matching conditions* that couples the local problems.

In this work we focus on the earliest and most elementary formulation termed *Schwarz hybrid formulation* (Mathew, 2008, Section 1.2) based on overlapping decomposition and is applicable to a wide class of self-adjoint and coercive elliptic equations. Given an overlapping decomposition, $\partial \Omega_k$ can be decomposed into two disjoint parts. One (possibly empty) part $\Gamma_k = \partial \Omega_k \cap \partial \Omega$ is located in the boundary of $\Omega$ and the global boundary condition should be imposed. The other part $B_k = \partial \Omega_k \cap \Omega$ is a nonempty artificial boundary from the overlapping decomposition and a Dirichlet boundary condition from the coupling of local problems is imposed.

We refer to Mathew (2008) for a strict definition. As an illustrative example, assume we have an overlapping decomposition with $K = 2$ and consider as the original problem Laplace equation with mixed Dirichlet and Neumann boundary conditions. The following coupled system of two local PDEs is a Schwarz hybrid formulation of the original problem and solving the original equation is equivalent to solving this coupled system.

$$
\begin{array}{llll}
\Delta u_1 = 0 & \text{in } \Omega_1 & \Delta u_2 = 0 & \text{in } \Omega_2 \\
u_1 = u_2|_{\partial \Omega_1} & \text{on } \partial \Omega_1 \cap \Omega & u_2 = u_1|_{\partial \Omega_2} & \text{on } \partial \Omega_2 \cap \Omega \\
u_1 = u_D & \text{on } \partial \Omega_1 \cap \Gamma_D \quad \text{and} & u_2 = u_D & \text{on } \partial \Omega_2 \cap \Gamma_D \\
\dfrac{\partial u_1}{\partial n} = g & \text{on } \partial \Omega_1 \cap \Gamma_N & \dfrac{\partial u_2}{\partial n} = g & \text{on } \partial \Omega_2 \cap \Gamma_N
\end{array}
$$

Based on the Schwarz hybrid formulation, there are various iterative schemes with different parallelism and convergence rate. In the subsequent discussion, our focus is primarily on introducing the *additive Schwarz methods (ASM)*. The ASM is a highly parallel algorithm (Mathew, 2008) in solving the coupled system from Schwarz hybrid formulation. We briefly introduce ASM with finite element methods and refer to Gander et al. (2008) and Mathew (2008) for details.

Assume that under weak formulation of Eq. 1 and finite element space $V$, Eq. 1 has the form $Au = f$ where $A$ is the stiffness matrix. Given an overlapping decomposition $\{\Omega_i\}_{k=1}^K$ compatible with the finite element space on $\Omega$, we have $V = \sum_{k=1}^K V_k$ as sum of local finite element subspaces $V_k$ on $\Omega_k$ and we can define local stiffness matrices $A_k : V_k \to V_k$, *restriction operators* $\{R_k\}_{k=1}^K$ restricting $V$ to $V_k$ and *extension operators* $\{R_k^\mathsf{T}\}_{k=1}^K$ extending $V_k$ to $V$ by zeros extension. We then define operators $P_k : V \to V$ by $P_k = R_k^\mathsf{T} A_k^{-1} R_k A$. *Additive Schwarz operator* is then defined as the

sum $P_{\text{ad}} = \sum_{k=1}^{K} P_k$. This operator can be show to be self-adjoint and coercive and we have the following equivalence.

$$Au = f \iff P_{\text{ad}}u = \sum_{k=1}^{K} R_k^{\mathsf{T}} A_k^{-1} R_k f \tag{4}$$

We note that the right hand side of Eq. 4 is a preconditioned version of the left hand side. The Richardson iteration for this preconditioned problem has the following form.

$$u^{n+1} = u^n + \tau \sum_{k=1}^{K} R_k^{\mathsf{T}} A_k^{-1} R_k (f - Au^n) \tag{5}$$

In the composite operator $R_k^{\mathsf{T}} A_k^{-1} R_k$, the operator $R_k$ first restrict a function to $\Omega_k$, $A_k^{-1}$ solve the local problem and $R_k^{\mathsf{T}}$ extend the local solution to $\Omega$. This iterative process can be shown to converge by estimating bound on condition number of $P_{\text{ad}}$ under mild assumptions on equation and decomposition.

## B    REVISIT ON OPERATOR LEARNING

The goal of operator learning is to learn a mapping $\mathcal{G} : \mathcal{A} \to \mathcal{U}$ between two infinitely dimensional spaces (Kovachki et al., 2023). When applied to PDEs, $\mathcal{U}$ is the solution space of a PDE and $\mathcal{A}$ is the space of functions that determine a unique solution of a PDE. Examples of $\mathcal{A}$ are coefficient functions or boundary/initial conditions that defines the PDE and parameters that determine the geometry of domain.

In our study, we decompose any domain into subdomains each of which lives in a distinguished class of basic shapes $\mathcal{P}$. We assume all shapes in $\mathcal{P}$ have Lipschitz boundary and are uniformly bounded, i.e., they are all bounded by a ball $D \subseteq \mathbb{R}^n$. We are interested in solving boundary value problems in Eq. 1 in any domain $\Omega \in \mathcal{P}$ with any appropriate boundary condition. We thus separate geometry and boundary conditions from other inputs and represent the input function space of the operator as $\mathcal{A} = \mathcal{P} \times H^k(D) \times \mathcal{H}$ where $H^k(D)$ is the Sobolev space $W^{k,2}(D)$. The space $\mathcal{P} \times H^k(D)$ represents the geometry of the domain together with boundary/initial conditions, $\mathcal{H}$ represents any other input functions such as coefficient function field or source term in the PDE. The neural operator thus approximates the following mapping. Note that in the case of time dependent problem, the space $H^k(D)$ represents the space of initial condition together with *time varying* boundary condition and the solution space $\mathcal{U}$ represents a time series up to some time span. The solution operator $\mathcal{G}$ thus has the following form.

$$\mathcal{G} : \mathcal{P} \times H^k(D) \times \mathcal{H} \to \mathcal{U} \tag{6}$$

For learning the operator, we assume $\mathcal{P}$, $H^k(D)$ and $\mathcal{H}$ are probability spaces and thus we can sample observations from $\mathcal{A}$. In practice, we randomly sample geometry from $\mathcal{P}$ and random boundary conditions are imposed, then a solution is generated from a numerical solver to get solutions. It is important to highlight that, unlike the usual setting for neural operators, there is significant variation in the shape of input domains.

## C    PROOF OF THEOREM 1

**Theorem.** *Assume the operator $\mathcal{L}$ in Eq. 1 is self-adjoint and coercive elliptic (Mathew, 2008). Let $u^n$ and $\tilde{u}^n$ denote the iterates of the classical additive Schwarz-Richardson iteration (Eq. 5) and SNI (Algorithm 1), respectively, with the same initialization $u^0 = \tilde{u}^0$.*

*Let $\mathcal{S}_k$ denote the exact local solution operator on $\Omega_k$, and define the learned (pre/post-processed) local solver $\tilde{\mathcal{S}}_k \triangleq \tilde{T}_k \circ \mathcal{G}^{\dagger} \circ T_k$. Assume the classical iteration mapping is contractive in a norm $\| \cdot \|$ with rate $0 < \rho < 1$, and the uniform local error bound $\|\tilde{\mathcal{S}}_k(\cdot) - \mathcal{S}_k(\cdot)\| \leq c$ holds for all $k$. Let $t$ be the maximum number of subdomains covering any degree of freedom. Then we have:*

- ***Convergence:** SNI converges to a fixed point $\tilde{u}^*$.*

- **_Error bound:_** _for all $n$, $\|\tilde{u}^n - u^n\| \leq c'$, and $\|\tilde{u}^* - u^*\| \leq c'$, where one may take $c' = \frac{\tau t}{1-\rho} c$._

*Proof.* (1) Write the classical and learned iterations in the same additive Schwarz form:

$$u^{n+1} = u^n + \tau \sum_{k=1}^{K} R_k^{\intercal} \big( \mathcal{S}_k(u^n) - R_k u^n \big),$$

$$ildeu^{n+1} = \tilde{u}^n + \tau \sum_{k=1}^{K} R_k^{\intercal} \big( \tilde{\mathcal{S}}_k(\tilde{u}^n) - R_k \tilde{u}^n \big),$$

(7)

where $\tilde{\mathcal{S}}_k \triangleq \tilde{T}_k \circ \mathcal{G}^{\dagger} \circ T_k$.

(2) Let $e^n \triangleq \tilde{u}^n - u^n$. Subtracting the two updates in Eq. 7 yields

$$e^{n+1} = \big( F(\tilde{u}^n) - F(u^n) \big) + \tau \sum_{k=1}^{K} R_k^{\intercal} \big( \tilde{\mathcal{S}}_k(\tilde{u}^n) - \mathcal{S}_k(\tilde{u}^n) \big),$$

where $F(u) \triangleq u + \tau \sum_{k=1}^{K} R_k^{\intercal}(\mathcal{S}_k(u) - R_k u)$ is the classical iteration mapping. By the contractivity assumption of the classical iteration, $\|F(\tilde{u}^n) - F(u^n)\| \leq \rho\|e^n\|$. Moreover, using the uniform local approximation error bound and the overlap factor $t$ (the maximum number of subdomains covering any degree of freedom), we have

$$\left\| \tau \sum_{k=1}^{K} R_k^{\intercal} \big( \tilde{\mathcal{S}}_k(\tilde{u}^n) - \mathcal{S}_k(\tilde{u}^n) \big) \right\| \leq \tau t c.$$

Therefore,

$$\|e^{n+1}\| \leq \rho\|e^n\| + \tau t c.$$

Iterating the inequality gives $\|e^n\| \leq \frac{1-\rho^n}{1-\rho} \tau t c$. Taking $c' \triangleq \frac{\tau t c}{1-\rho}$ yields $\|\tilde{u}^n - u^n\| \leq c'$ for all $n$.

(3) Since $u^n$ converges by assumption, it is Cauchy. For any $m > n$, we have

$$\|\tilde{u}^m - \tilde{u}^n\| \leq \|u^m - u^n\| + \|\tilde{u}^m - u^m\| + \|\tilde{u}^n - u^n\| \leq \|u^m - u^n\| + 2c'.$$

Hence $\tilde{u}^n$ is also Cauchy and thus converges to some $\tilde{u}^*$. Passing to the limit in $\|\tilde{u}^n - u^n\| \leq c'$ yields $\|\tilde{u}^* - u^*\| \leq c'$, completing the proof. $\qquad\square$

If we apply matrix form of neural operator, namely, the neural operator aims to approximate $\{A_k^{-1}\}_{k=1}^{K}$ and assume, then we can have the following result:

**Corollary 1.** *Consider the exact operator $A_k^{-1}$ and inexact neural operator $\tilde{A}_k^{-1}$, $k = 1, \cdots, K$. Let $u^n$ and $\tilde{u}^n$ represent the solutions updated by $A_k^{-1}$ and $\tilde{A}_k^{-1}$ respectively at the $n$-th step, where the updating rule is given by Eq. 5 with $\tau = 1$ and both sharing the same initialization. Suppose that $\| A_k^{-1} - \tilde{A}_k^{-1} \| < c$, for $k = 1, \cdots, K$, and $\rho(I - MA) < 1$, where $M = \sum_{k=1}^{K} R_k^{\intercal} \tilde{A}_k^{-1} R_k$, then we have:*

- *Convergence: the algorithm converges to a fixed point;*

- *Error bound: there exists a constant $c_1(c)$ such that $\| \tilde{u}^n - u^n \| < \frac{c_1}{\|I-MA\|}$;*

- *Condition number: $\kappa(MA) \leq \min(t(K+1), 1 + \max_k \frac{H_k}{d})$,*

*where $t$, $K$, $H_k$ and $d$ denote the maximal number of overlapping subdomains, the number of subdomains, the diameter of $k$-th subdomain, and the number of extensions, respectively.*

Note that the condition $\rho(I - MA) < 1$ is generally challenging to satisfy. To address this issue, we employ the Richardson iteration trick (Richardson, 1911) in order to ensure the convergence of the proposed algorithm (Algorithm 1).

## D    TIME-DEPENDENT PROBLEMS.

We consider the time-dependent PDE with the following form:

$$
\begin{aligned}
u_t - \mathcal{L}u &= f && \text{in } \Omega \times [0, T] \\
u(x, t) &= u_D(x, t) && \text{on } \partial\Omega \times [0, T] \\
u(x, 0) &= u_0(x) && \text{on } \Omega \times \{0\}
\end{aligned}
\tag{8}
$$

where $\mathcal{L}$ is again self-adjoint and coercive elliptic operator. The additive Schwarz method can be naturally extended to a *space-time* additive Schwarz method (Li and Cai, 2015) by considering a decomposition of the space-time domain $\Omega \times [0, T]$ by taking the product of overlapping decomposition of $\Omega$ and $[0, T]$ respectively. The space-time domain decomposition has the form $\Omega_i \times [t_{j-1} - \delta_T, t_j + \delta_T]$ where $\delta_T$ is the temporal depth and represent overlap in time domain. Once such a decomposition is constructed, the same additive Schwarz method can be applied to the space-time decomposition to get a global solution on the space-time domain, allowing parallel iteration in both space and time domain. Local problems for the above decomposition are again of the form in Eq. 8.

In our implementation on heat equation, we discretize the time domain with a fixed time step $t_s$, fix a rollout length of $k$ and train a neural operator to map initial and boundary conditions to time series for the $k$ steps at $t = 0, t_s, \cdots, (k-1)t_s$. More precisely, the neural operator is trained to map $u_D$ and $u_0$ to time series of the form $u(x, 0), u(x, t_s), \cdots, u(x, (k-1)t_s)$.

## E    SYMMETRIES OF PDEs

The *symmetry group* of a general partial differential operator $\mathcal{L}$ refers to a set of transformations that map a solution to another solution, forming a mathematical group. *Lie point symmetry* is a subgroup of the symmetry group that has a Lie group structure and acts on functions *pointwise* as transformations on coordinates and function values (Brandstetter et al., 2022). In this work, we will in addition be concerned with not just a single operator $\mathcal{L}$, but a family of operators depending on various coefficient fields (e.g., Darcy flow) and various boundary/initial conditions. Symmetries have to be properly extended to these input functions so that a solution with an input function is transformed to another solution with a different input function.

Leveraging these symmetries allows for the generation of an infinite number of new solutions based on a given solution. The idea of utilizing these symmetries as a data augmentation technique for operator learning was initially introduced in Brandstetter et al. (2022). However, we apply these data augmentation to solutions on basic shapes in training local operator and this usage of symmetries echos a point mentioned in Brandstetter et al. (2022, Section 3.2) where the authors point out that these data augmentation can be applied on local patches of solutions instead of the solution on the entire domain.

There is another direct usage of symmetries in our framework. Instead of incorporating symmetries as a form of data augmentation in training time, one can directly apply transformations to input and output of a neural operator during inference time. We implement these transformations as preprocessing and postprocessing steps in the inference pipeline. We summarize the symmetries of each PDE applied in our implementation in Table 3. Normalizations applied as preprocessing and postprocessing for each of the equations are summarized in Table 6.

## F    TIME COMPLEXITY

### F.1    EMPIRICAL TIME COMPLEXITY

We provide empirical results on runtime for our main results. We discuss how better initialization can accelerate the whole iterative process in the next point.

We first provide empirical runtime on a single sample for each stationary problem and each domain reported in our main result. We use the following metrics:

- Time to convergence (TTC).

| Equation | Lap2d-D | Lap2d-M | Darcy2d | Heat2d | NonLap2d |
|---|---|---|---|---|---|
| Spatial Shift | $(x_1, x_2) \to (x_1 + t_1, x_2 + t_2)$ | | | | |
| Spatial Rotation | $(x_1, x_2) \to (x_1 \cos\theta - x_2 \sin\theta, x_1 \sin\theta + x_2 \cos\theta)$ | | | | |
| Spatial Scaling | $(x_1, x_2) \to (sx_1, sx_2)$ | | | | |
| Spatial Scaling | – | $u \to su$ 
 $u_D \to su_D$ 
 $g \to g$ | $u \to s^2 u$ 
 $u_D \to s^2 u_D$ 
 $a(x) \to a(x)$ 
 $f(x) \to f(x)$ | $u \to u$ 
 $u_0 \to u_0$ 
 $u_D \to u_D$ 
 $\alpha \to s^2\alpha$ | – |
| Value Shift | $u \to u + t$ 
 $u_D \to u_D + t$ | | | | – |
| Value Scaling | $u \to su$ 
 $u_D \to su_D$ | | | | – |
| Value Scaling | – | $g \to sg$ | – | $u_0 \to su_0$ | – |

Table 3: Symmetries of various PDEs applied in our implementation.

- Time to 15/10/5% relative $l_2$ error following the practice in Mao et al. (2024).

| Equations | Domains | TTC(s) | TT15%(s) | TT10%(s) | TT5%(s) |
|---|---|---|---|---|---|
| Laplace2d-Dirichlet | A | 100 | 40 | 50 | 70 |
| Laplace2d-Dirichlet | B | 269 | 107 | 137 | 194 |
| Laplace2d-Dirichlet | C | 28 | 8 | 11 | 17 |
| Laplace2d-Mixed | A | 162 | 82 | 103 | 137 |
| Laplace2d-Mixed | B | 714 | 511 | 620 | - |
| Laplace2d-Mixed | C | 68 | 43 | 52 | - |
| Darcy2d | A | 84 | 37 | 54 | - |
| Darcy2d | B | 247 | 144 | 176 | - |
| Darcy2d | C | 26 | 12 | 15 | - |

Table 4: Empirical runtime for different equations and domains

Factors that affect the runtime are:

1. Type of equations. We observe that Laplace2d-Mixed takes longer on all domains. We also observe that the existence of Neumann boundary condition leads to a larger range of function values for the solution of Laplace2d-Mixed. This leads to more iterations steps required to reach convergence.

2. Number of subdomains $K$ and step size $\tau$. In the above table, domain B takes longer for all equations because it has 40 subdomains compared to 20 for A and C. A large number of subdomains leads to more time consumption for an iteration. We illustrated how choice of $\tau$ affects the number of iterations to convergence in section 4.3 of our paper.

3. Local operator architecture. While GNOT gets better results in accuracy, a drawback of transformer-based methods is that they are usually slower than FNO (Hao et al., 2023).

4. Initialization. This is discussed in the next point.

We note that our implementation is not optimized to fully parallelize the iterative process; for example, the normalization process is not parallelized in our implementation.

Numerical solvers are very fast in generating solutions for the domains we tested on and we do not expect our approach to be faster than these highly optimized numerical solvers on these (still) simple domains. As a reference, generating a solution for Laplace2d-Dirichlet on domain A using classical FEM solution takes $6.15 \times 10^{-4}$ seconds and performing a GNOT inference on the same domain takes $1.26 \times 10^{-2}$ seconds. We can see that even classical numerical solver is faster than GNOT inference. However, DDMs are a conventional approach implemented in commercial software

designed to solve PDEs on large-scale and complicated domains. We replace the local FEM solver in DDMs by a data-driven neural operator and thus expect our approach to show superiority when the problem domain is large and complicated.

## F.2 Acceleration through Better Initialization

We discuss how to accelerate the iterative process by starting with a better initialization. In the original implementation, we always start with a zero solution in the interior of the domain. To accelerate the process, we initialize with solutions from GNOT direct inference and find that it considerably saves our time. We report the time consumption on Laplace2d-Dirichlet using the same metrics as the previous table. The only difference is in initialization.

| Equations | Domains | TTC(s) | TT15%(s) | TT10%(s) | TT5%(s) |
|---|---|---|---|---|---|
| Laplace2d-Dirichlet | A | 28 | 8 | 11 | 17 |
| Laplace2d-Dirichlet | B | 70 | 3 | 6 | 21 |
| Laplace2d-Dirichlet | C | 20 | 1 | 2 | 7 |

Table 5: Runtime results with improved initialization for Laplace2d-Dirichlet equations

However, coming up with a better initialization is not trivial and can be an interesting future work.

## G Discussions

**Message passing in DDMs.** In our framework, we solve a coupled system of local problems by an iterative algorithm SNI. Through iteratively solving local problems based on boundary values from the last iteration and thus from neighboring subdomains, SNI is essentially performing message passing between subdomains. This message passing operation may be implemented in other forms, e.g., through a graph neural network.

**Higher-dimensional PDEs.** Our framework can be extended to higher-dimensional cases as long as basic shapes and corresponding solutions can be properly generated. For 3-d problems, one potential selection of basic shapes is the class of polytopes.

The fundamental "local-to-global" principle of our method remains unchanged: decompose a complex 3D domain into smaller, canonical 3D subdomains, solve locally with a neural operator, and iteratively stitch the solutions. However, implementation faces additional challenges in 3D. For example, generating building blocks in 3D. One way is to use polyhedrons as building blocks in 3D. Another choice is to generate building blocks from "growing" tetrahedra in 3D. Both can be very interesting ways to explore. In summary, extending our method to 3D is a challenging but highly promising research trajectory. It is not a simple "plug-and-play" extension but requires careful research at the intersection of computational geometry, operator learning, and high-performance computing.

**Other formulations of DDMs.** Schwarz hybrid formulation discussed in this work is one of the most elementary formulation in DDMs. There are many other more advanced DDMs (Mathew, 2008). *Steklov-Poincaré framework* is based on non-overlapping decomposition and *transmission condition* as coupling condition for local problems. *Langrange multiplier framework* leads to the well-known *FETI* method and is also based on non-overlapping decomposition.

**Other types of PDEs.** The additive Schwarz method in classical DDMs works for self-adjoint and coercive elliptic equations. Non-self-adjoint elliptic equations, parabolic equations, saddle-point problems and non-linear equations requires separate treatment. Addressing these cases presents challenges in both training the local operator and designing the iterative algorithm.

**Future Works.** Based on the above discussion, there are many potential directions for future works. First, it would be interesting to implement this framework using a message-passing framework instead of an iterative algorithm to accelerate the convergence. Second, extending our framework to address higher-dimensional problems is important, particularly since industrial problems often involve 3-d simulation. Third, more advanced DDMs such as Neumann-Neumann, BDDC and FETI (Mathew, 2008) may also be explored. Lastly, other types of PDEs such as saddle point problems and non-linear equations such as Navier-Stokes equation is out of the scope of our current work, and present unique

challenge. Tackling these challenges requires not only expertise on operator learning, but also deep understanding of PDEs themselves. We speculate that it would be fruitful to combine rich literatures of DDMs with operator learning.

# H    EXPERIMENTS

## H.1    DATASETS

Here we introduce more details of our datasets in both training and testing stage. For training data, we generate random simple polygons with $3 \leq n \leq 12$ vertices within $[-0.5, 0.5]^2$ and create uniform mesh using Gmsh (Geuzaine and Remacle, 2008). We prepare a separate dataset for validation during training. For testing data, we generate the three domains depicted in Figure 2 together with mesh using the Gmsh UI. We argue that the complexity of geometric domains is fundamentally determined by their underlying topological and geometrical properties. Based on this intuition, we considered three domains of increasing complexity for evaluation: (1) Domain A: This domain is simply connected, representing the simplest class of geometries; (2) Domain B: This domain has two holes and is multiply connected, indicating a higher level of complexity compared to the simply connected Domain A; (3) Domain C: This domain has one hole with corners, further increasing the geometrical complexity compared to the previous two domains. Through a systematical evaluation across this spectrum of domains, from the simple geometry to more intricate multiply connected domains with holes and corners, we believe the results provide a comprehensive understanding of our framework's capabilities.

Once the geometries and meshes are created, we specify boundary/initial conditions for various equations and domains and generate solutions using FEniCSx (Baratta et al., 2023; Scroggs et al., 2022a;b), a popular open-source platform for solving PDEs with the finite element method (FEM). We adopt Lagrange element of order 1 (linear element) as our finite element space in generating boundary/initial conditions and solutions. Next we give details on how these boundary/initial conditions and solutions are generated for each type of PDEs. We also summarize these details in Table 7 and 8.

**Laplace2d-Dirichlet.** Laplace equation in 2d with pure Dirichlet boundary condition. The governing equation is

$$\begin{aligned} \Delta u &= 0 &&\text{in } \Omega \\ u &= u_D &&\text{on } \partial\Omega. \end{aligned} \tag{9}$$

For both training and testing data, we specify piecewise linear Dirichlet boundary condition with randomly generated values within $[0, 1]$ on boundary nodes.

**Laplace2d-Mixed.** Laplace equation in 2d with mixed Dirichlet and Neumann boundary condition on $\partial\Omega = \Gamma_D \cup \Gamma_N$. The governing equation is

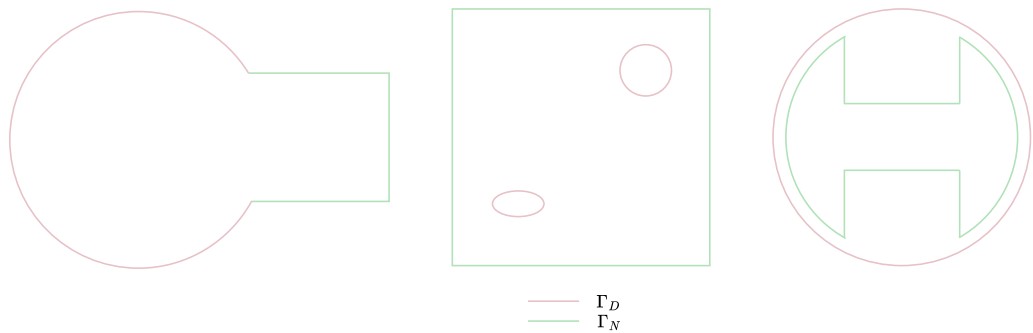

Figure 5: Illustration of mixed Dirichlet and Neumann boundaries for domain A, B, C in Laplace2d-Mixed.

$$\begin{aligned}
\Delta u &= 0 && \text{in } \Omega \\
u &= u_D && \text{on } \Gamma_D \\
\frac{\partial u}{\partial n} &= g && \text{on } \Gamma_N.
\end{aligned} \tag{10}$$

For training data, $20\%$ of the data have pure Dirichlet condition and the generation process is the same as in Laplace2d-Dirichlet. The rest $80\%$ of the training data have mixed boundary condition with non-empty connected Neumann boundary and Neumann boundary is randomly specified to be less than half of the entire boundary. Then a random number $r$ is sampled from $U[0.5, 1]$ to specify functional range for Dirichlet and Neumann boundaries as described next. Among data with non-empty Neumann boundary, $50\%$ have $u_D \in [0, r]$ and $g \in [0, 1]$ and the other $50\%$ have $u_D \in [0, 1]$ and $g \in [0, r]$.

For testing data, Dirichlet and Neumann boundary is specified for each of the domain A,B and C as shown in Figure 5. Boundary conditions $u_D$ and $g$ are both piecewise linear with randomly generated values within $[0, 1]$.

**Darcy2d.** Darcy flow in 2d with coefficient field $a(x)$, source term $f(x)$ and pure Dirichlet boundary condition. The governing equation is

$$\begin{aligned}
-\nabla(a(x)\nabla u) &= f && \text{in } \Omega \\
u &= u_D && \text{on } \partial\Omega.
\end{aligned} \tag{11}$$

For training data, Dirichlet boundary condition is specified with a random range $r \in [0.3, 1]$ and boundary values are generated as $u_D \in [0, r]$. The coefficient function $a(x)$ and source term $f(x)$ are specified as piecewise linear functions with randomly generated values within $[0, 1]$ on nodes.

For testing data, Dirichlet boundary condition, coefficient function and source term are all piecewise linear functions with randomly generated values within $[0, 1]$.

**Heat2d.** Time-dependent equation of heat conduction in 2d with coefficient $\alpha$ denoting the thermal diffusivity, time-varying boundary condition and initial condition. The governing equation is

$$\begin{aligned}
\frac{\partial u}{\partial t} &= \alpha\Delta u && \text{in } \Omega \times [0, T] \\
u(x, t) &= u_D(x, t) && \text{on } \partial\Omega \times [0, T] \\
u(x, 0) &= u_0(x) && \text{on } \Omega \times \{0\}.
\end{aligned} \tag{12}$$

For training data, we discretize the time domain with a fixed time step $t_s = 0.01$, generate piecewise linear initials and time-varying boundary conditions with values randomly generated within $[0, 1]$. $\alpha$ is a random number within $[0.8, 1]$. We adopt the backward Euler method (Langtangen and Logg, 2017) and generate a time series of 10 time steps. During training we separate these 10 time steps into 2 time series of 5 times steps and training the neural operator to predict 5 time steps.

For testing data, we fix $\alpha = 1$. Initial condition is piecewise linear with values randomly generated within $[0, 1]$. Boundary condition is specified to be constant over time and varied randomly within $[0, 1]$ across boundary nodes. We also adopt the backward Euler method and generate a time series of 50 time steps.

**NonlinearLaplace2d.** A nonlinear Laplace equation in 2d with pure Dirichlet boundary condition following an example in Langtangen and Logg (2017). The governing equation is

$$\begin{aligned}
\nabla \cdot ((u^2 + 1)\nabla u) &= 0 && \text{in } \Omega \\
u &= u_D && \text{on } \partial\Omega.
\end{aligned} \tag{13}$$

For both training and testing data, we specify piecewise linear Dirichlet boundary condition with randomly generated values within $[0, 1]$ on boundary nodes.

## H.2 Evaluation Protocol and Hyperparameters.

**Evaluation Protocol.** The evaluation metric we utilize is the mean $l_2$ relative error. Let $u_i, u_i' \in \mathbb{R}^n$ represent the ground truth solution and predicted solution for the $i$-th sample, respectively. Considering a dataset of size $D$, the mean $l_2$ relative error is computed as follows:

$$\varepsilon = \frac{1}{D} \sum_{i=1}^{D} \frac{\| u_i' - u_i \|_2}{\| u_i \|_2} \tag{14}$$

**Hyperparameters.** All experimental hyperparameters used in the paper are listed in Table 6. For data generation, the number of vertices of simple polygons are uniformly chosen between 3 to 12. And $a \times b$ in configurations denotes the generation of $b$ shapes, each having $a$ distinct boundary/initial conditions. For investigating data efficiency issue, we only vary the number of various shapes $b$ while keeping the number of random input functions per shape $a$ constant. For boundary condition imposition, we summarize the details in Table 7 and 8.

**Computing Resource.** We run our experiments on 1 Tesla V100 GPU.

| | | Lap2d-D | Lap2d-M | Darcy2d | Heat2d | NonlinearLap2d |
|---|---|---|---|---|---|---|
| **Data Generation** | Polygon | | | [3,12] | | |
| | Training Configuration | 10×2000 | 20×2000 | 10×4000 | 50×1600 | 10× 2000 |
| | Validation Configuration | 10×250 | 20×200 | 10×250 | 50×240 | 10× 2000 |
| | Testing Configuration | 100 | 100 | 100 | 10 | 100 |
| **Operator Learning** | GNOT | | | 1 expert and 3 layers of width 128 | | |
| | Optimization | | | Adam | | |
| | Learning rate | | | cycle learning rate strategy with 0.001 | | |
| | Epoch | 500 | 1000 | 500 | 200 | 500 |
| | Data Aug. | Rot. | Rot.+ Sca. [0.8,1] | No | Rot.+ Sca. [0.8,1] | Rot. |
| | Time steps | | − | | 5 | − |
| **Inference (SNI)** A | Partition $K$ | | 20 | | 20×16 | 20 |
| | Depth $d$ | | | 2 | | |
| | Temp. Depth $\delta_T$ | | − | | 1 | − |
| | Step size $\tau$ | | 0.04 | | 0.002125 | 0.04 |
| | Pre/Post-pro. | Spa. Shift+Scale Val. Shift+Scale | | Spa. Shift+ Scale | Spa. Shift+Scale Val. Shift+Scale | Spa. Shift |
| B | Partition $K$ | | 40 | | 40×16 | 40 |
| | Depth $d$ | | | 2 | | |
| | Temp. Depth $\delta_T$ | | − | | 1 | − |
| | Step size $\tau$ | | 0.024 | | 0.0014625 | 0.024 |
| | Pre/Post-pro. | Spa. Shift+Scale Val. Shift+Scale | | Spa. Shift+ Scale | Spa. Shift+Scale Val. Shift+Scale | Spa.Shift |
| C | Partition $K$ | | 20 | | 20× 16 | 20 |
| | Depth $d$ | | | 2 | | |
| | Temp. Depth $\delta_T$ | | − | | 1 | − |
| | Step size $\tau$ | | 0.04 | | 0.002125 | 0.04 |
| | Pre/Post-pro. | Spa. Shift+Scale Val. Shift+Scale | | Spa. Shift+ Scale | Spa. Shift+Scale Val. Shift+Scale | Spa. Shift |

Table 6: Key hyperparameters of main experiments. Configuration under Data Generation is specified as (number of random input functions per shape) × (number of various shapes). Partition $K$ for Heat2d is specified as (number of spatial partition) × (number of temporal partition).

| PDE | Description |
|---|---|
| Lap2d-D | range of boundary condition: $U[0,1]$ |
| Lap2d-M | 20% pure Dirichlet condition: range $U[0,1]$
40% mixed boundary condition with $\Gamma_D/\partial\Omega \sim U[0.5,1]$
    range of Dirichlet: $U[0,r]$ where $r \sim U[0.5,1]$, range of Neumann:$U[0,1]$
40% mixed boundary condition with $\Gamma_D/\partial\Omega \sim U[0.5,1]$
    range of Dirichlet: $U[0,1]$, range of Neumann: $U[0,r]$ where $r \sim U[0.5,1]$ |
| Darcy2d | range of boundary conditions: $U[0,r]$ where $r \sim U[0.3,1]$,
range of $a(x)$ and $f$: $U[0,1]$ |
| Heat2d | range of initial/boundary condition: $U[0,1]$ , $\alpha \sim U[0.8,1]$ |
| NonlinearLap2d | range of boundary condition: $U[0,1]$ |

Table 7: Details of boundary/initial condition and input function generation in training data.

| PDE | Description |
|---|---|
| Lap2d-D | range of boundary condition: $U[0,1]$ |
| Lap2d-M | $\Gamma_D$ and $\Gamma_N$ as in Figure 5
range of Dirichlet: $U[0,1]$, range of Neumann: $U[0,1]$ |
| Darcy2d | range of boundary conditions: $U[0,1]$
range of $a(x)$ and $f(x)$: $U[0,1]$ |
| Heat2d | $\alpha = 1$
range of boundary/initial condition: $U[0,1]$
boundary condition do not vary with time |
| NonlinearLap2d | range of boundary condition: $U[0,1]$ |

Table 8: Details of boundary/initial condition and input function generation in testing data.

## H.3 VISUALIZATION OF BASIC SHAPES AND DOMAIN DECOMPOSITION.

**Visualization of basic shapes.** We provide examples of generated basic shapes for training in Figure 6.

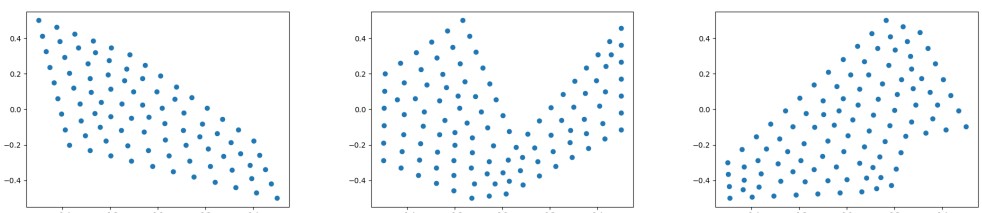

Figure 6: Examples of basic shapes.

**Visualization of decomposed domains: A, B, and C** We provide the visualization of decomposed domains A, B, C in Figure 7. Please be noted that this is just a rough visualization in that we do not correctly plot the overlapping part of subdomains. So this is only for an intuitive understanding of the decomposed domain.

## H.4 OTHER SUPPLEMENTARY RESULTS.

**Data Efficiency.** The results for data efficiency on Laplace2d-Mixed and Darcy2d are shown in Figure 8. The average performance of SNI is better than GNOT on all of three domains, while the margins between the two methods on domains A and B are not statistically significant due to the high variance in the $l_2$ relative errors of SNI on these two domains. GNOT struggles in the generalization to domain C, while SNI can still handle it with a good performance.

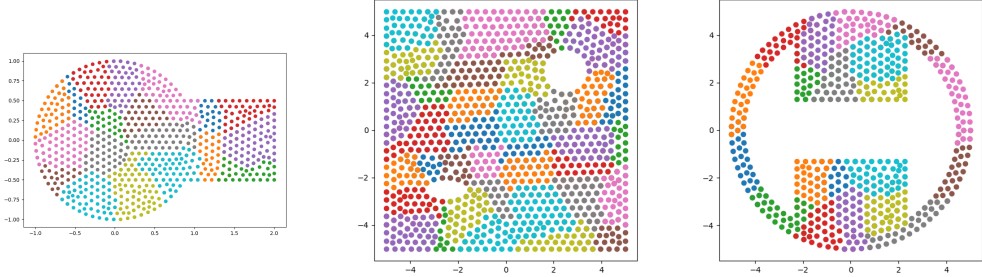

Figure 7: Visualization of decomposed domains A, B and C.

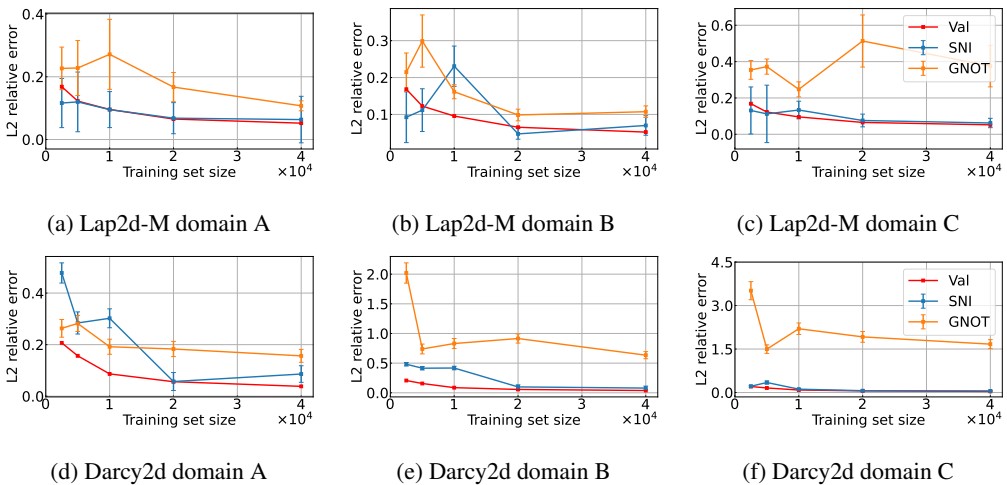

(a) Lap2d-M domain A

(b) Lap2d-M domain B

(c) Lap2d-M domain C

(d) Darcy2d domain A

(e) Darcy2d domain B

(f) Darcy2d domain C

Figure 8: Comparison between the $l_2$ relative errors from SNI (blue), GNOT direct inference (orange) and validation (red) on Laplace2d-Mixed and Darcy2d upon three domains (A, B and C) with different numbers of training samples.

**Irregular and More Complicated Domain: Dolphin-shape Domain.** We provide the result of Laplace2d-Dirichlet on a dolphin-shape domain in Table 9 which has more complex boundary. The mesh and decomposed domain is illustrated in Figure 9. We can see that on this irregular and more complicated domain, the SNI with GNOT still gets reasonably good result. The error is relatively higher than that of simpler domains in Table 1. This error gap can be caused by the gap between training and testing shape distribution.

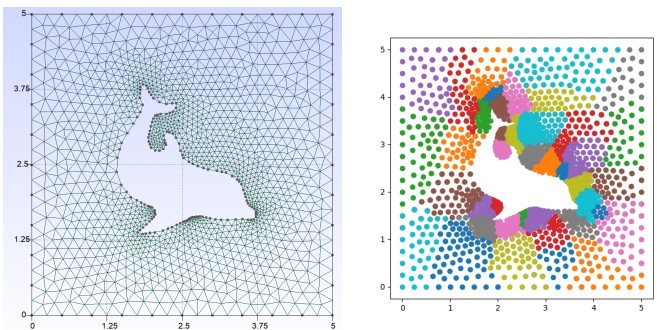

Figure 9: Visualization of dolphin-shape domain.

| Equation | Domain | SNI with GNOT (%) |
|---|---|---|
| Laplace2d-Dirichlet | Dolphin | 4.5± 0.9 |
|  | Disk | 2.1±0.5 |

Table 9: Laplace2d-Dirichlet on dolphin-shape and disk domains.

| Equation | Domain | $val_{GNOT}$ | SNI with GNOT | $val_{Geo-FNO}$ | SNI with Geo-FNO |
|---|---|---|---|---|---|
| Laplace2d-Dirichlet | A | 2.5 | 2.2±0.6 | 5.6 | 9±3 |
|  | B |  | 2.1±0.4 |  | 14±1 |
|  | C |  | 2.1±0.9 |  | 12±2 |
| Darcy2d | A | 3.8 | 9±2 | 6.4 | 11±2 |
|  | B |  | 8±2 |  | 15.7±0.9 |
|  | C |  | 5.4±0.6 |  | 20±2 |

Table 10: SNI with GNOT or Geo-FNO as different choices of local operator for Laplace2d-Dirichlet and Darcy2d. Validation errors are provided for reference.

**Simple Domain: Disk.** We provide the result of Laplace2d-Dirichlet on a simple disk domain in Table 9. This result is comparable to these in Table 1.

**Comparison with Graph-based Neural Networks (GNN).** We provide the result of direct inference with MeshGraphNets (Pfaff et al., 2020). We train the GNN on our Laplace2d training data and evaluate it on the domains A. We get a relative $l_2$ error of 11.5%. We find that GNN does provide better generalization across different domains compared to GNOT and it can be potentially used to accelerate our iterative algorithm in our future work.

**Choice of neural operator architecture.** We provide results of SNI with Geo-FNO Li et al. (2023) on Laplace2d-Dirichlet and Darcy2d in Table 10.

**Solution and Error Visualization.** We provide visualization for stationary problems in Section 4. We visualize ground-truth solution from testing data, absolute error from SNI and GNOT direct inference in Figure 10, 11 and 12. We also provide the error visualization in Figure 14 along with the decomposed domain to understand where the error is located.

## I BROADER IMPACTS

First, the proposed framework holds the potential to serve as an alternative to conventional PDE solving tools. Through its ability to address challenges related to geometry-generalization and data efficiency, the framework offers advantages that can significantly improve the efficiency of PDE solving. This improvement can have a positive impact on various industries, including engineering, physics, and finance, where PDEs are extensively employed for modeling and simulation purposes.

Second, the proposed three-level hierarchy for PDE generalization provides researchers with valuable directions for future exploration in neural operator research. This hierarchical structure offers a framework to systematically address the challenges associated with generalizing neural operators to new geometries and PDEs. By considering these three levels, researchers can focus on developing techniques and methodologies that improve the adaptability, flexibility, and scalability of neural operators. Furthermore, current operator learning methods in the neural operator field are predominantly driven by data and do not adequately consider the underlying PDE information. In our research, we introduce domain decomposition into the neural operator domain to tackle the issue of geometric generalization, incorporating traditional PDE approaches. This research direction presents significant potential for further investigation.

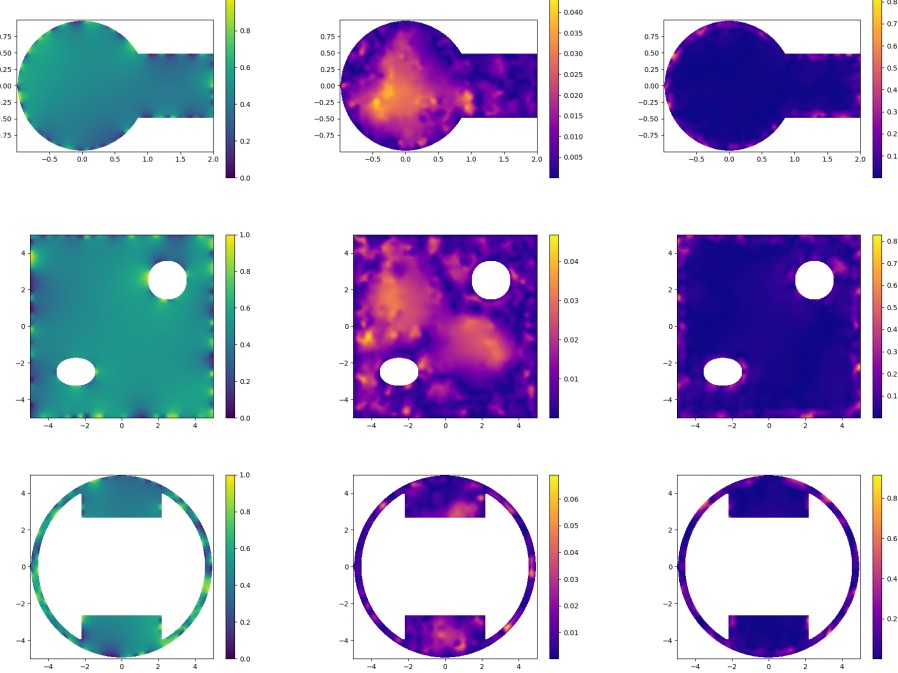

Figure 10: Visualization of test dataset of Laplace2d-Dirichlet on domain A, B and C. The three columns from left to right display the ground-truth solution, absolute error from SNI and GNOT direct inference.

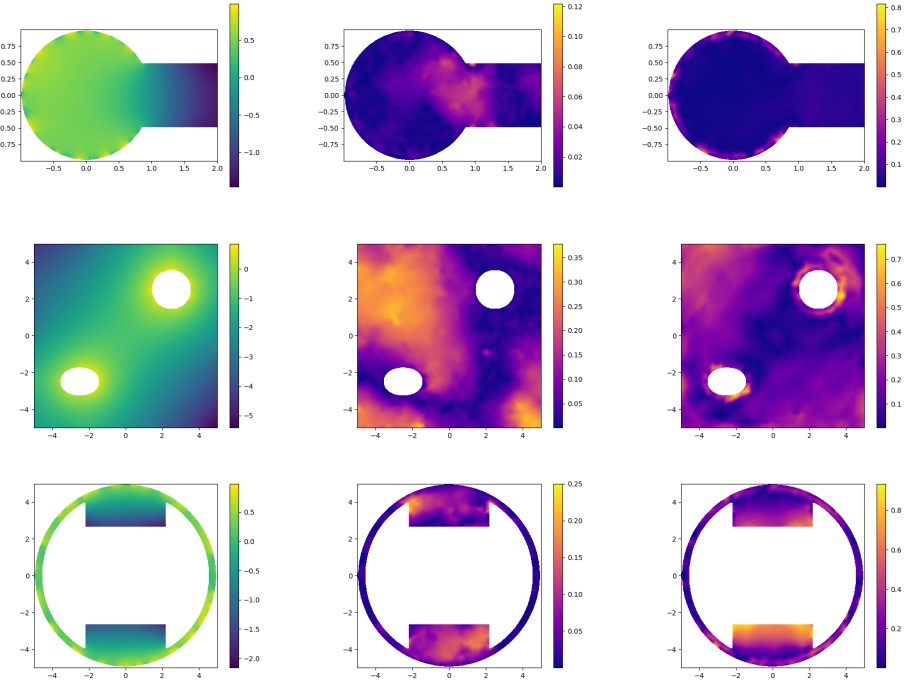

Figure 11: Visualization of test dataset of Laplace2d-Mixed on domain A, B and C. The three columns from left to right display the ground-truth solution, absolute error from SNI and GNOT direct inference.

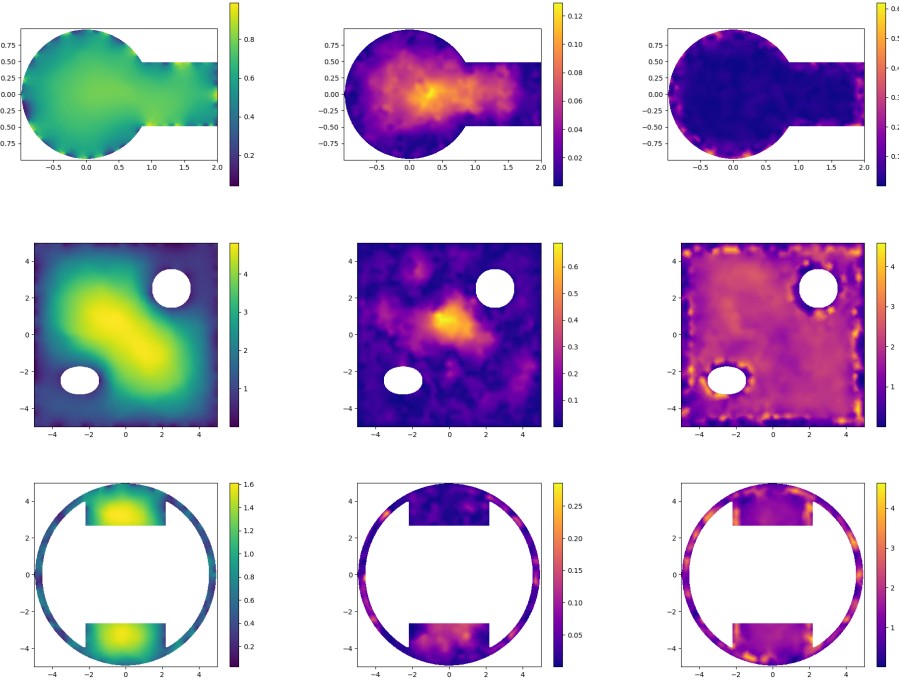

Figure 12: Visualization of test dataset of Darcy2d on domain A, B and C. The three columns from left to right display the ground-truth solution, absolute error from SNI and GNOT direct inference.

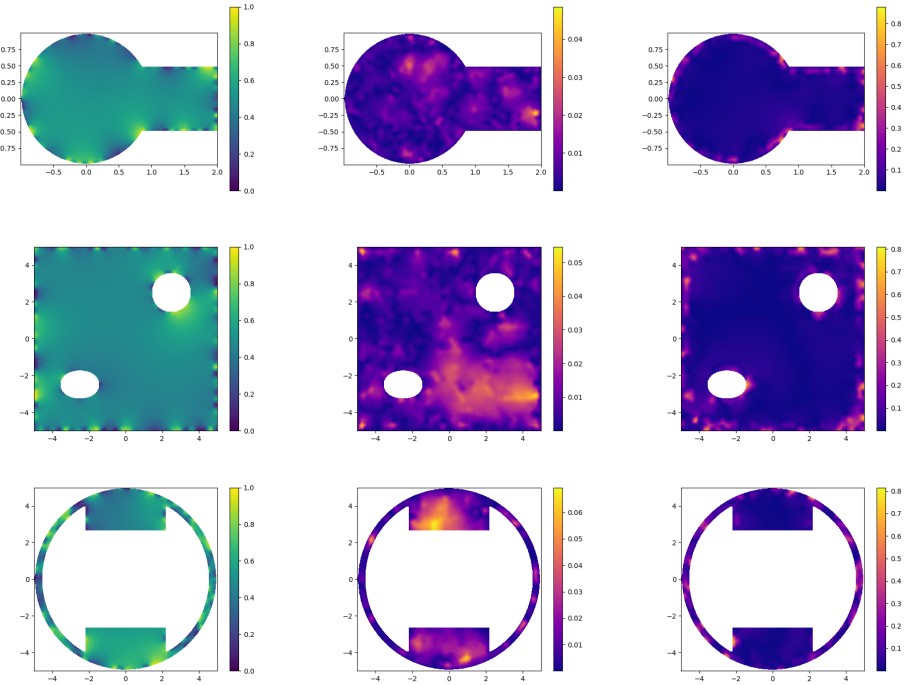

Figure 13: Visualization of test dataset of NonlinearPoisson2d on domain A, B and C. The three columns from left to right display the ground-truth solution, absolute error from SNI and GNOT direct inference.

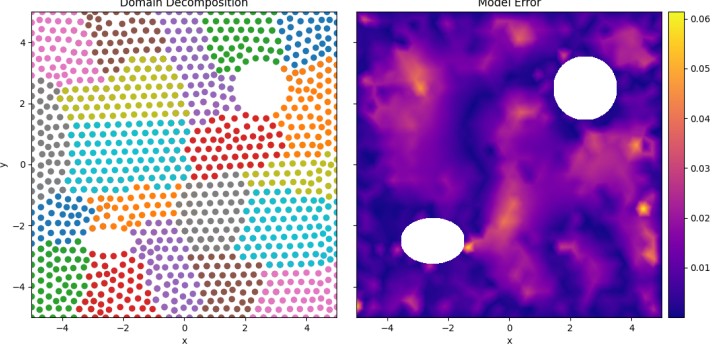

Figure 14: Visualization of error distribution together with decomposed domain for Laplace2d-Dirichlet on domain B.

