# OpenReview forum: "Operator Learning with Domain Decomposition for Geometry Generalization in PDE Solving"
_ICLR.cc/2026/Conference — ICLR 2026 Poster_

### Official Review · Reviewer_kNbL · 2025-10-26

**Soundness:** 3
**Presentation:** 3
**Contribution:** 3
**Rating:** 4
**Confidence:** 3

**Summary:**

The authors propose a new framework for training neural operators to generalize to unseen geometries at inference with domain decomposition. Instead of generating a variety of complex geometries and train a neural operator such that it can generalize to an unseen complex geometry at test time, the authors instead propose to train a neural operator on a diverse number of simple geometries that can be seen as a decomposition of a complex geometry. At inference, they introduce the SNI algorithm, that decompose a complex unseen geometry into a set of simpler shapes, that can be solved iteratively.

**Strengths:**

I really like the tackled problem and find it very interesting and well motivated. Data driven approaches are known to be robust to the geometries observed in the training distribution, but they naturally tend to fail when tested on geometries that shift from the training distribution. Decomposing a complex geometry into a set of very simple shapes (that thus are more aligned with the training distribution) is an interesting approach to generalize any neural operator to arbitrary geometry.

Experiments successfully demonstrate that the framework proposed help to boost performance of neural operators on unseen geometries via geometry decomposition. The framework is general and ca be applied to any neural operator that accepts arbitrary mesh points.

Ablations are provided to see how sensible is the method with respect to the number of sub-domains K, data augmentation, etc.

**Weaknesses:**

While the motivation is clear, one of the potential benefit of neural operators is to overcome the computational cost of traditional methods. Decomposing the domain into sub-domains naturally increase the computational cost of data-driven approaches. Ablations provided in appendix seem to show that the computational time needed is high for very accurate predictions. It would be nice to compare this with the time needed for solving the PDE with a numerical method, and put it as limitations based on the results.

The experiments have been tested on relatively simple and a limited number of geometries. It would be nice to test the method on more challenging geometries and see how well it performs with respect to the number of sub-domains. If the geometry is much more complex, we can imagine that it would need much more sub-domains to recover the simple geometries seen during training.

I think the paper would clearly benefit from a related work that mentions existing methods for geometry generalization. The method should be compared to existing works when possible.

**Questions:**

Could you provide runtime comparisons? It would be helpful to provide actual runtime comparisons: (a) pure neural operator direct inference, (b) SNI (with partition & iteration), (c) classical PDE solver.

The method has been tested under specific boundary conditions and in a specific setting. Could you elaborate a bit on how general this method for PDE modeling? How could it be applied to more challenging physical systems?

**Details Of Ethics Concerns:**

No concerns

---

> ### Author Response · Authors · 2025-11-21
>
> > While the motivation is clear, one of the potential benefit of neural operators is to overcome the computational cost of traditional methods. Decomposing the domain into sub-domains naturally increase the computational cost of data-driven approaches. > Ablations provided in appendix seem to show that the computational time needed is high for very accurate predictions. It would be nice to compare this with the time needed for solving the PDE with a numerical method, and put it as limitations based on the results.
>
> Please see below
>
>
> > The experiments have been tested on relatively simple and a limited number of geometries. It would be nice to test the method on more challenging geometries and see how well it performs with respect to the number of sub-domains. If the geometry is much more complex, we can imagine that it would need much more sub-domains to recover the simple geometries seen during training.
>
> In the Appendix I.4 we provide an additional geometries which is a dolphin-shaped domain with irregular geometries and showcase that our algorithm also perform well on it. The reviewer is right in that the number of subdomains to cover the domain is 40 and is significantly larger than the domain A,B and C.
>
> > I think the paper would clearly benefit from a related work that mentions existing methods for geometry generalization. The method should be compared to existing works when possible.
>
> We have included more works in the 5.Related work in the revised version. There have been existing work dealing with geometry generalization using PINN as local solver such as xPINN or cPINN. The SNAP-DDM in "Towards general neural surrogate solvers with specialized neural accelerators. ICML’24" tries to accelerate domain decomposition methods with neural operator. However it works with uniform grid instead of arbitrary mesh and the domain used there is specifically designed. We find there is a lack of work on domain generalization with neural operators that can work on arbitrary domain with potentially non-uniform grid.
>
> > Could you provide runtime comparisons? It would be helpful to provide actual runtime comparisons: (a) pure neural operator direct inference, (b) SNI (with partition & iteration), (c) classical PDE solver.
>
> Generating a solution for Laplace2d-Dirichlet on domain A using classical FEM solution takes 6.15e-4 seconds and performing a GNOT inference on the same domain takes 1.26e-2 seconds. We can see that even classical numerical solver is faster than GNOT inference. However, DDMs are a conventional approach implemented in commercial software designed to solve PDEs on large-scale and complicated domains. We replace the local FEM solver in DDMs by a data-driven neural operator and thus expect our approach to show superiority when the problem domain is large and complicated.
>
>
> > The method has been tested under specific boundary conditions and in a specific setting. Could you elaborate a bit on how general this method for PDE modeling? How could it be applied to more challenging physical systems?
>
>
> We thank the reviewer for this insightful question regarding the generality of our framework. Our method is designed as a general paradigm for solving PDEs, and its application in the paper on specific equations is a proof of concept. The core idea—decomposing a complex global problem into manageable local problems solvable by a learned operator—is widely applicable.
> The generality stems from three key aspects:
>
> • Architecture-Agnostic Design: Our framework is orthogonal to the neural operator choice. As shown in our experiments, it works with both GNOT and Geo-FNO. Any future, more powerful neural operator can be seamlessly integrated as the local solver.
>
> • Separation of Concerns: We decouple geometric complexity (handled by domain decomposition) from physics learning (handled by the local operator). As long as the PDE's fundamental physics can be captured locally, the global solution can be assembled for any domain geometry.
>
> • Robust Theoretical Foundation: Our work builds on rigorous Domain Decomposition Methods (DDMs), which have a proven track record across a vast spectrum of problems far beyond the elliptic and parabolic equations shown here.
> Our framework fully supports Dirichlet, Neumann, and mixed Dirichlet-Neumann boundary conditions on the global domain. The decomposition and inference algorithm are designed to handle the complex mixed BCs that arise on local subdomains during the iterative process.
>
> Our current implementation has not been tested with Robin boundary conditions. While the domain decomposition theory can be extended to include Robin BCs, incorporating them would require ensuring our local neural operator is trained on such conditions and that the conditions in the inference algorithm are appropriately defined. This represents a valuable direction for future extension.

---

### Official Review · Reviewer_bCzG · 2025-10-29

**Soundness:** 2
**Presentation:** 2
**Contribution:** 2
**Rating:** 4
**Confidence:** 3

**Summary:**

This paper introduced an operator learning strategy based on domain decomposition to handle complex geometries. The paper proposes to use data augmentation to improve the learning of the PDE on simple geometries. The solving method is based on domain-decomposition and locally uses the neural operator. Finally the method reconstruct dynamics on complex geometries by merging sub-solutions, using a SNI algorithm. The paper proposes a theoretical study of the algorithm as well as some experiments.

**Strengths:**

- The proposed paper is well-writen and carrefully introduces a lot of concepts.
- A theoretical analysis is proposed as well as experimental evaluations.

**Weaknesses:**

- Scope of applicability: My concerns about applicability relies in the fact that the problem formulation states that PDE are tested in parabolic equations (with time). Is it a limitation of the method? Why is the method limited to such PDEs?
- Missing references: a lot a references dealing with complex geometries and generalizations are missing, see eg [1-5]
- No experiments on public datasets: I think evaluating the proposed method on well-known dataset helps the reader positioning the method with respect to existing baselines. As an example, some dataset considering irregular/complex geometries are in [6-7].
- While the paper is very rich, several key components of the method are detailed in appendices, making the reading as is hard to understand (eg algorithm).
- Inference time and scaling : I have some concerns regarding these 2 aspects of the method, see questions.

### References
[1] Operator Learning with Neural Fields: Tackling PDEs on General Geometries, Louis Serrano, Lise Le Boudec, Armand Kassaï Koupaï, Thomas X Wang, Yuan Yin, Jean-Noël Vittaut, Patrick Gallinari, 2023

[2] Transolver: A Fast Transformer Solver for PDEs on General Geometries, Haixu Wu, Huakun Luo, Haowen Wang, Jianmin Wang, Mingsheng Long, 2024

[3] A.D.Jagtap, G.E.Karniadakis, Extended Physics-Informed Neural Networks (XPINNs): A Generalized Space-Time Domain Decomposition Based Deep Learning Framework for Nonlinear Partial Differential Equations, Commun. Comput. Phys., Vol.28, No.5, 2002-2041, 2020.

[4] K. Shukla, A.D. Jagtap, G.E. Karniadakis, Parallel Physics-Informed Neural Networks via Domain Decomposition, Journal of Computational Physics 447, 110683, (2021).

[5] AROMA: Preserving Spatial Structure for Latent PDE Modeling with Local Neural Fields, Louis Serrano, Thomas X Wang, Etienne Le Naour, Jean-Noël Vittaut, Patrick Gallinari, 2024

[6] Learning Mesh-Based Simulation with Graph Networks, Tobias Pfaff, Meire Fortunato, Alvaro Sanchez-Gonzalez, Peter W. Battaglia, 2020

[7] A comprehensive and fair comparison of two neural operators (with practical extensions) based on FAIR data, Lu Lu, Xuhui Meng, Shengze Cai, Zhiping Mao, Somdatta Goswami, Zhongqiang Zhang, George Em Karniadakis, 2021

**Questions:**

### Questions
-	Could you compare the proposed method to more recent baselines ? consider public/standard datasets of the litterature? Do you plan on releasing the dataset for future work?
-	Line 093-095, you mentionned that the method is restricted to parabolic equations. Could you detail about this ? What are limitations in terms of PDE equations/BCs (also reference to lines 198-199)?
-	Could you provide some insight on how to chose the hyper parameters mentioned in sections 3.1 (ie n?, K, d?) It is specified that hyper parameters have to be carefully chosen, why ? Doesn’t the method converge in these cases ? It does not seems to be very sensitive to hp in terms of final performance in the ablation on figure 4.
-	Could you detail about the results on Heat2d – B ? While the method seems coherent and robust in all scenarios, on this geometry however, results drops x2.
-	How does the method behaves on simple geometries ?
-  could you elaborate on the condition required for the algorithm to converge? Line 160 refers to Appendix A, but Appendix A finally states that convergence depends on the PDE and decomposition.
- Have you studied where are located the errors? Is there a higher (or lower) error around the boundary of the local shapes?
- How does the method scales in term of memory consumption with respect to the number of points ? Would it be tractable for 3D PDEs ? Industrial complex geometries ?
-	Have you compared inference time with respect to baselines ? I saw the method’s inference time in the appendices, but I could not find baselines’.
- Regarding the theoretical section: I think I am not qualified to review such theoretical results, but I have question regarding the applicability of the proof: When are the assumptions met? For example, could you cite some PDE respecting the hypothesis on L?

### Minor comments :
-	Legend of figure3 : it it hard to understand what validation refers to, I think it is detailed in the last sentence of description? just adding a (validation) at the end would remove any questioning.
-	Figure 4 : I think that log scales for errors would be more appropriate for these figures. As is, it is hard to visualize the differences between all ablation (at least the 2 firsts).

---

> ### Author Response · Authors · 2025-11-21
>
> > Scope of applicability: My concerns about applicability relies in the fact that the problem formulation states that PDE are tested in parabolic equations (with time). Is it a limitation of the method? Why is the method limited to such PDEs?
>
> Please see below
>
> > Missing references: a lot a references dealing with complex geometries and generalizations are missing, see eg [1-5]
>
> Thanks for pointing out these references, we have included all of them in 5. Related work in the revised manuscript .
>
> > No experiments on public datasets: I think evaluating the proposed method on well-known dataset helps the reader positioning the method with respect to existing baselines. As an example, some dataset considering irregular/complex geometries are in [6-7].
>
> Please see below
>
> > While the paper is very rich, several key components of the method are detailed in appendices, making the reading as is hard to understand (eg algorithm).
>
> We put the algorithm in the appendix due to limited space. We will put the algorithm in the main text in the revised version
>
> > Inference time and scaling : I have some concerns regarding these 2 aspects of the method, see questions.
>
> Please see below
>
> > Could you compare the proposed method to more recent baselines ? consider public/standard datasets of the litterature? Do you plan on releasing the dataset for future work?
>
> Since we are proposing a new task which is the geometry generalization problem, we find it hard to find available public datasets that can meet our needs. For example we require training data to consist of solution on basic shapes which is not available in any public datasets. In addition, we require the testing domain to have mesh with them for domain decomposition and that is also not available in public dataset. These are our major motivation to generate our own dataset. In the Appendix I.4 we provide an additional geometries which is a dolphin-shaped domain with irregular geometries and showcase that our algorithm also perform well on it.
> We will release our dataset together with data generation pipeline which is complete transparent since they are all generated using open-source framework FEniCS.
>
>
> > Line 093-095, you mentionned that the method is restricted to parabolic equations. Could you detail about this ? What are limitations in terms of PDE equations/BCs (also reference to lines 198-199)?
>
> We thank the reviewer for this question, which allows us to precisely clarify the scope and current limitations of our method. The statement regarding time-dependent parabolic equations can be a bit misleading in the problem formulation. What we really mean is that in the implementation of time-dependent PDEs, we only implement the time-dependent parabolic equation. We will delete the term "parabolic" in the revised version. This is not a limitation of our method.
> Regarding our theoretical result, our framework and the core Schwarz iteration can be proved to converge for second-order, self-adjoint, and coercive elliptic operators. This class includes the Laplace and Darcy equations presented in our work. However, empirically this method can be applied to other PDEs. We have included one non-linear PDE-Non-linear Laplace equation in 2D-in our experiments, and our method works quite well.

---

> ### Author Response · Authors · 2025-11-21
>
> > Could you provide some insight on how to chose the hyper parameters mentioned in sections 3.1 (ie n?, K, d?) It is specified that hyper parameters have to be carefully chosen, why ? Doesn’t the method converge in these cases ? It does not seems to be very sensitive to hp in terms of final performance in the ablation on figure 4.
> > could you elaborate on the condition required for the algorithm to converge? Line 160 refers to Appendix A, but Appendix A finally states that convergence depends on the PDE and decomposition.
>
> We thank the reviewer for this excellent question, which pushes us to clarify a crucial point. The reviewer is correct that our main text and appendix point to the properties of the PDE and decomposition as the key to convergence. We apologize for the lack of clarity and are happy to provide a more precise elaboration here.
> The convergence of the classical additive Schwarz method (and by extension, our SNI algorithm under the error bound assumption in Theorem 1) is rigorously guaranteed under the following set of sufficient conditions:
> 1. PDE Properties: The operator $\mathcal{L}$ must be linear, self-adjoint, and coercive (i.e., elliptic in the sense of leading to a symmetric positive definite (SPD) bilinear form in the weak formulation). This is a standard assumption for the convergence theory of additive Schwarz
> 2. Decomposition Properties: The overlapping domain decomposition must satisfy two key geometric criteria:
> ○ Finite Covering: Each point in the global domain $\Omega$ can belong to at most $T$ overlapping subdomains, where $T$ is a finite number.
> ○ Sufficient Overlap: The width of the overlap region between subdomains, $d$, must be sufficient relative to the mesh size. In practice, even a small overlap (e.g., one or two layers of mesh elements) is often enough to ensure convergence.
> When these conditions are met, one can prove that the condition number of the preconditioned system $P_{ad}$ is bounded by a constant that depends on $T$ and $d$ but is independent of the number of subdomains  and the global problem size. This independence is the key to the numerical scalability of the method. A bounded condition number directly implies that the Richardson iteration will converge.
> In summary, the algorithm is guaranteed to converge for linear, self-adjoint, coercive PDEs (like the Laplace and Darcy equations in our experiments) when using a decomposition with finite overlap and a finite covering number.
> The hyperparameter K is also important for our method to work reasonably well. The reason is that we have to split a large domain into small subdomains that fall in the shapes distribution of our generated basic shapes. When K is too small, the subdomains may not be simple polygons which is out of our shape distribution. When K is too large, it takes more iterations to converge as illustrated by Figure 4
>
> > Could you detail about the results on Heat2d – B ? While the method seems coherent and robust in all scenarios, on this geometry however, results drops x2.
>
> This example is by itself harder than others. Heat2d is a temporal example and is actually 2+1 D in comparison with other 2D exmaples, so it may be expected that the method might have a larger error on harder examples.
>
> > How does the method behaves on simple geometries ?
>
> We provide the result on a simple disk domain in the revised version. Please look at Appendix I.4 for the result. It is working as good as on domain A, B and C
>
> > Have you studied where are located the errors? Is there a higher (or lower) error around the boundary of the local shapes?
>
> We provide a visualization of the error along with the decomposed domain in Appendix I.4 in the revised version. Our observation is that the error can be relatively high in some small subdomains instead of around the boundary only. We suspect that is caused by the gap between the distribution of training and testing shapes.

---

> ### Author Response · Authors · 2025-11-21
>
> >  How does the method scales in term of memory consumption with respect to the number of points ? Would it be tractable for 3D PDEs ? Industrial complex geometries ?
>
> We thank the reviewer for raising these critical questions on scalability. Our method is inherently designed for efficient memory scaling, which we highlight as a key advantage over global neural operator approaches.
> The memory footprint of our framework is fundamentally different from and superior to global approaches:
>
> • The dominant memory cost comes from evaluating the neural operator on individual subdomains.
>
> • Crucially, the size of each local problem is bounded by the largest subdomain size M and does not grow with the global problem size N.
>
> • Most importantly, these subdomain solves are highly parallelizable. This means that in practice, the peak memory consumption is dominated by the memory required for the single largest subdomain, O(M). This is because all K subdomain solves can be distributed across available computational resources (e.g., GPUs).
>
> This structure stands in stark contrast to a single global neural operator, which must load the entire domain (of size N) into memory at once, often exhibiting super-linear memory growth (e.g., O(N²) for attention) and creating a fundamental bottleneck.
> In summary, by decomposing the domain, we replace a single memory-intensive global problem with many small, independent problems that can be solved in parallel. This not only alleviates memory constraints but also enables the solution of problems that are intractable for monolithic neural operators.
>
> >  Have you compared inference time with respect to baselines ? I saw the method’s inference time in the appendices, but I could not find baselines’.
>
> Generating a solution for Laplace2d-Dirichlet on domain A using classical FEM solution takes 6.15e-4 seconds and performing a GNOT inference on the same domain takes 1.26e-2 seconds. We can see that even classical numerical solver is faster than GNOT inference. However, DDMs are a conventional approach implemented in commercial software designed to solve PDEs on large-scale and complicated domains. We replace the local FEM solver in DDMs by a data-driven neural operator and thus expect our approach to show superiority when the problem domain is large and complicated.
>
>
> >  Regarding the theoretical section: I think I am not qualified to review such theoretical results, but I have question regarding the applicability of the proof: When are the assumptions met? For example, could you cite some PDE respecting the hypothesis on L?
>
> We thank the reviewer for this crucial question regarding the applicability of our theoretical assumptions. It is a very valid point, and we are happy to clarify the scope of our theoretical results.
> The core assumptions in Theorem 1 are that the partial differential operator $\mathcal{L}$ is self-adjoint and coercive. These are standard and well-understood concepts in the theory of partial differential equations and are the foundational assumptions for the convergence of classical additive Schwarz methods, upon which our work builds. A wide range of canonical PDEs fall into this category. Prominent examples include:
> 1. The Laplace/Poisson Equation: $-\Delta u = f$. This is the prototypical example of a self-adjoint and coercive operator. It models phenomena from electrostatics to steady-state heat conduction.
> 2. The Helmholtz Equation (in certain regimes): $-\Delta u - k^2 u = f$. While the Helmholtz equation can be challenging for high wave numbers k, it is self-adjoint, and coercivity can be established under certain conditions, often involving a shift or in bounded domains.
> 3. General Second-Order Linear Elliptic Equations: in the form $-\nabla \cdot (A(\mathbf{x})\nabla u) + c(\mathbf{x})u = f$, where the coefficient matrix A(x) is symmetric, positive definite, and c(x) ≥ 0**. This is a very broad class that includes, for instance, the equations governing
>
>  ○ Isotropic/Anisotropic Diffusion: where A(x) is a diffusion tensor.
>
>  ○ Linear Elasticity (in its elliptic equilibrium form).
>
> These operators are central to many fields in physics and engineering. The self-adjoint property typically arises from symmetric physical laws and variational principles (e.g., energy minimization), while coercivity ensures the well-posedness of the boundary value problem.

---

> > ### Comment · Reviewer_bCzG · 2025-11-27
> > **Answer to author's rebuttal**
> >
> > I thank the authors for their detailed answers to my numerous questions. I greatly appreciate the new results on complex geometries, the visualizations, and the modifications brought to the paper.
> > - Regarding the experiments on new geometries: My understanding is that, when comparing the results on the simple circle shape to those provided in Table 1, the method generalizes quite well to more complex geometries, as the error rates remain close (2% versus 2-5 % in average ?).
> > - Regarding the error visualization: I find these results interesting. It seems that errors are not located at the same positions for GNOT and SNI. Additionally, it appears that for some PDEs, the GNOT model predicts a trivial 0 solution (eg figure 11, shapes B and C). Could you please comment on this? Is this a failure mode of the baseline or a visualization artifact?
> >
> > I also thank the authors for the additional details on the method's complexity and the theoretical analysis.
> > With these additional explanations and details, I will increase my score to 6.
> > I strongly encourage the authors to include the explanations provided in their rebuttal within the final paper, particularly regarding the scaling properties of their method, as this will be very useful for future readers and reproducibility.

---

> > > ### Author Response · Authors · 2025-11-28
> > >
> > > > Regarding the experiments on new geometries: My understanding is that, when comparing the results on the simple circle shape to those provided in Table 1, the method generalizes quite well to more complex geometries, as the error rates remain close (2% versus 2-5 % in average ?).
> > >
> > > Yes this is the right observation. On very simple shape like circle, the method generalizes better
> > >
> > > > Regarding the error visualization: I find these results interesting. It seems that errors are not located at the same positions for GNOT and SNI. Additionally, it appears that for some PDEs, the GNOT model predicts a trivial 0 solution (eg figure 11, shapes B and C). Could you please comment on this? Is this a failure mode of the baseline or a visualization artifact?
> > >
> > > - The reviewer has the right observation that the errors for GNOT and SNI are not located at the same position. Inference with GNOT is performed once on the entire domain and it is very smooth. Inference with SNI employs an iterative algorithm on subdomains. That is the reason of the more "rough" looking of the error from SNI.
> > > - The last column in Figure 11 is the visualization of the absolute error between ground-truth and GNOT. We can see from the color bar that the maximum error can be up to around 5 for B and 2.5 for C. The second column in Figure 11 is the visualization of the absolute error between ground-truth and SNI. We can see from the color bar that the maximum error is around 0.35 for B and 0.25 for C. Since the error for SNI and GNOT are not in the same range, we use a different range for the color bar in the second column and third column of Figure 11. While the error of GNOT has a more "smooth" looking, it is significantly larger than the error of SNI. The location that appears to be a zero for GNOT is not actually zero due to this visualization artifact from the color bar
> > >
> > > > I also thank the authors for the additional details on the method's complexity and the theoretical analysis. With these additional explanations and details, I will increase my score to 6. I strongly encourage the authors to include the explanations provided in their rebuttal within the final paper, particularly regarding the scaling properties of their method, as this will be very useful for future readers and reproducibility.
> > >
> > > We thank reviewer for your valuable questions that allow us to clarify some important points in our work. We also thank reviewer for raising the rating. We will include these explanations in the final paper to help reader better understand our method and limitations.

---

> > > > ### Comment · Reviewer_bCzG · 2025-11-28
> > > > **Discussion on the error visualization**
> > > >
> > > > Thanks again for your quick answer. Regarding the second point, what I meant here is rather that it looks like the GNOT baseline predicts a 0 trivial solution as output? Do you have any insight explaining this behavior?
> > > > For instance, in figure 11b, the ground truth has -5 value at 2 of the edges of the square (top left and bottom right), and the associated error is 5, while around the 2 obstacles, the ground truth value is 0 and the associate error of GNOT is 0. This is what made me wondering about the prediction provided by the baseline.

---

> > > > > ### Author Response · Authors · 2025-11-28
> > > > >
> > > > > > Thanks again for your quick answer. Regarding the second point, what I meant here is rather that it looks like the GNOT baseline predicts a 0 trivial solution as output? Do you have any insight explaining this behavior? For instance, in figure 11b, the ground truth has -5 value at 2 of the edges of the square (top left and bottom right), and the associated error is 5, while around the 2 obstacles, the ground truth value is 0 and the associate error of GNOT is 0. This is what made me wondering about the prediction provided by the baseline.
> > > > >
> > > > > Thanks for clarifying the question. We checked the prediction from GNOT and see that the absolute value is relatively small (around 0) compared to the range of the color bar. but not strictly 0. We suspect that GNOT has this behavior because in the training dataset where the shapes are small, solution with mixed Dirichlet and Neumann boundary conditions tends to have a small absolute value because of the limited spatial range for value of the harmonic function to "transition" from Dirichlet boundary (within [0,1]) to Neumann boundary (gradient in normal direction within [0,1]). However for domains with larger distance between Dirichlet and Neumann boundary, we can see much bigger deviation in function value between Dirichlet boundary and Neumann boundary.
> > > > > This can also be seen by looking at the ground-truth solution of Figure 11( the first column). For domain B, the function value is smaller on points of the Neumann boundary(the outer boundary of the square) that is farther from the Dirichlet boundary( the two holes, inner boundary). For domain C, since the disance between every point of the Neuman boundary(the inner contour) and the Dirichlet boundary(the outer contour) is smaller, such transition is not as dramatic as for domain B.

---

### Official Review · Reviewer_1tUX · 2025-10-30

**Soundness:** 3
**Presentation:** 3
**Contribution:** 3
**Rating:** 8
**Confidence:** 4

**Summary:**

This paper addresses generalization to novel geometries in neural operator-based PDE solving. The approach builds on domain decomposition theory, specifically proposing a neural version of Schwarz inference with theoretical guarantees. The authors train models (GNOT) on simple geometries, then apply them at inference to complex domains that can be decomposed into subdomains. The iterative method that the authors propose consistently outperforms the vanilla direct inference.

**Strengths:**

- As seen in Table 1, the method obtains excellent results on elliptic problems and on the temporal Heat equation.
- Figure 4 is particularly enlightening and shows that the error consistently decreases as the number of iterations grows, which highlights the stability of the method.
- The choice of shapes at test time is interesting and appears non trivial, even if they are in 2d.

**Weaknesses:**

- The authors don't provide in the main paper illustrations for the simpler domains that they choose during training.
- The method assumes that a partitioning is already available at test time, which suggests that the method relies on some preprocessing. This needs to come with further explanations and would help a reader less familiar with domain decomposition method how it works in practice.
- Figure 4 shows that in some cases the method requires up to 5000 iterations to converge. How much inference time overhead does this represent compared to the direct inference.
- The role of the data augmentation is not very clear: Table 2 shows that this has a little effect on the generalization performance and I would therefore insist less on this aspect than on the analysis of the core of the method.

**Questions:**

- How do you embed the boundary conditions in the GNOT architecture? Does it differ from the way you do it with GeoFNO? Could this explain the performance gap? Is your method really model agnostic or do you assume specific capabilities (e.g. boundary conditions encoding) in the neural operator?
- Could you provide visualisations for some training domain examples?
- Could this method be extended to 3d domains?

---

> ### Author Response · Authors · 2025-11-21
>
> > The authors don't provide in the main paper illustrations for the simpler domains that they choose during training.
>
> We thank the reviewer for this valuable suggestion. Due to space constraints, these examples were not included in the main text. Following the reviewer's suggestion, we have added a figure in the Appendix I.3 showcasing a collection of these randomly generated simple polygons.
>
> > The method assumes that a partitioning is already available at test time, which suggests that the method relies on some preprocessing. This needs to come with further explanations and would help a reader less familiar with domain decomposition method how it works in practice.
>
> We appreciate the reviewer for highlighting this point. Our inference algorithm, SNI, relies on an overlapping domain decomposition of the target geometry as a preprocessing step. This is a standard and highly automated procedure in classical Domain Decomposition Methods (DDMs). We adopt the mature graph partitioning tool METIS to achieve automated decomposition. We described the procedure in Section 3.3: Decomposition into overlapping subdomains with an illustration in Fig 1b. The steps are as follows:
>
> 1. Triangulation: The input domain $\Omega$ is discretized into a mesh $T_h(\Omega)$, which is a standard prerequisite for both numerical solvers and many neural operator architectures.
>
> 2. Non-overlapping Partition: A graph is constructed based on the mesh connectivity. A graph partitioner (METIS) is then used to partition this graph into K non-overlapping, connected subgraphs.
>
> 3. Create Overlap: Each non-overlapping subdomain is extended d times by iteratively including all immediate neighboring mesh nodes. This generates the final set of overlapping subdomains ${Ω_k}_{k=1}^K$.
>
> This automated process ensures that the resulting subdomains resemble the simple polygons in our predefined family P, allowing the pre-trained local neural operator to be directly applied. We will enhance the explanation of this pipeline and reference Figure 1b, which provides an intuitive illustration, to make it more accessible.
>
> > Figure 4 shows that in some cases the method requires up to 5000 iterations to converge. How much inference time overhead does this represent compared to the direct inference.
>
> This is a very good point to discuss. While one inference just takes less than 10e-2 s to complete, thousands of iterations takes tens of seconds to complete. However, the code is not optimized and one cannot expect that the computations overhead is minimized. For example, the normalization process is not parallelized well. We expect the computation overhead to be reduced siginificantly after a careful improvement in parallelization and optimization of code. Please see Appendix G for empirical time complexity.
>
> > The role of the data augmentation is not very clear: Table 2 shows that this has a little effect on the generalization performance and I would therefore insist less on this aspect than on the analysis of the core of the method.
>
> We thank the reviewer for this suggestion. We agree that the core contribution of our work is the local-to-global framework and the SNI algorithm, and we should focus on the narrative accordingly.
>
> The purpose of including data augmentation (specifically, Lie point symmetry) was to explore whether it could further improve the robustness of the local operator, especially when facing a wide range of geometries and boundary conditions during inference. As the reviewer notes and as we honestly report in Table 2, its impact is not always significant and can even be detrimental if applied improperly (e.g., with an overly aggressive scaling range). That is largely due to the inability of the neural operator to fit very aggressive data augmentation. In practice, we encounter nan in training the local operator when the data augmentation is too aggressive.
>
> Following the reviewer's advice, we will retain the results but frame them more cautiously, as we currently do in lines 432-433: "Hence, it is important to apply data augmentation with caution and consider its suitability for different types of PDEs." This refocusing will help readers more clearly grasp the primary strengths of our proposed framework.

---

> ### Author Response · Authors · 2025-11-21
>
> > How do you embed the boundary conditions in the GNOT architecture? Does it differ from the way you do it with GeoFNO? Could this explain the performance gap? Is your method really model agnostic or do you assume specific capabilities (e.g. boundary conditions encoding) in the neural operator?
>
> In the GNOT architecture, one can embed the boundary condition naturally as an input function, as GNOT suports the input of disretized functions. In GeoFNO, since the shape of input must match that of the output, there is no such freedom of functional input. We frame the input as one single point cloud, with the boundary values equal to the input boundary conditions and other values equal to zero. The performance gap is mainly due to the information processing in two different models. In GNOT, the attention between every single input point cloud and input functions is evaluated, which learns the relationship more directly. In GeoFNO, the input representing boundary condition suffers from a sparse information density. This input point cloud with the majority of points as zero goes through a Fourier transform and results in a function with mostly lowe frequency signals. This signal is difficult to be mapped back to the solution point cloud. Our method is model agnostic, but is dependent on the ability of model to solve boundary value problems on irregular mesh.
>
>
> > Could you provide visualisations for some training domain examples?
>
> Some visualizations of training domain examples can be found in Appendix I.3 of the revised manuscript.
>
> > Could this method be extended to 3d domains?
>
> We thank the reviewer for this question regarding the tractability of our method for 3D PDEs. The short answer is yes, our framework is in principle directly applicable to 3D, but its practical implementation presents distinct and non-trivial challenges that form the core of our ongoing and future work.
>
> The fundamental "local-to-global" principle of our method remains unchanged: decompose a complex 3D domain into smaller, canonical 3D subdomains, solve locally with a neural operator, and iteratively stitch the solutions. However, implementation faces additional challenges in 3D. For example, generating building blocks in 3D. One way is to use polyhedrons as building blocks in 3D. Another choice is to generate building blocks from "growing" tetrahedra in 3D. Both can be very interesting ways to explore.
> In summary, extending our method to 3D is a challenging but highly promising research trajectory. It is not a simple "plug-and-play" extension but requires careful research at the intersection of computational geometry, operator learning, and high-performance computing.

---

### Official Review · Reviewer_HhMK · 2025-10-31

**Soundness:** 3
**Presentation:** 2
**Contribution:** 2
**Rating:** 4
**Confidence:** 3

**Summary:**

This paper proposes operator learning with domain decomposition, a novel framework designed to improve the geometry generalization and data efficiency of neural operators for solving PDEs. By integrating domain decomposition methods with neural operator learning, the authors train local neural operators on randomly generated basic shapes and then use an iterative inference algorithm called Schwarz Neural Inference to stitch local solutions into a global one for arbitrary geometries. The paper provides theoretical guarantees on convergence and error bounds and demonstrates the method’s strong performance across a variety of linear and nonlinear PDEs.

**Strengths:**

- Develop a local-to-global framework with strong theoretical grounding that seamlessly integrates domain decomposition and neural operator learning, offering flexibility to support multiple neural operator architectures for solving PDEs on complex geometries.

- Demonstrates robust geometry generalization, effectively overcoming one of the major limitations of existing neural operator methods.

- Presents comprehensive experiments encompassing both linear and nonlinear, as well as steady and transient, PDEs.

**Weaknesses:**

- The paper does not justify that domains A, B, and C adequately represent the full diversity of 2D geometries. Moreover, all experiments are limited to 2D cases, leaving the scalability and applicability to 3D geometries unverified. At least one experiment on a 3D domain would have strengthened the validation of the proposed framework.

- Since the computational cost heavily depends on the number of iterations required by the iterative inference process, efficiency can become a concern. Moreover, the number of subdomains Kmay depend on the number of vertices and edges, especially when field variations within the domain are significant. However, the authors treat Kas independent of the number of vertices and edges when estimating the time complexity, which can be misleading. This assumption suggests a linear relationship with the number of vertices, while in practice, the computational complexity may scale multiplicatively with both the number of vertices and the number of iterations.

- Although the proposed framework represents the computational domain as a graph, the paper does not include any comparison or discussion with graph-based neural networks or their variants (e.g., [1, 2, 3]), which could provide relevant baselines or complementary perspectives.

[1] Neural Operator: Graph Kernel Network for Partial Differential Equation, 2020.

[2] Learning Mesh-Based Simulation with Graph Networks, ICLR, 2021.

[3] HAMLET: Graph Transformer Neural Operator for Partial Differential Equations, ICML, 2024.

**Questions:**

See the weakness

---

> ### Author Response · Authors · 2025-11-21
>
> > The paper does not justify that domains A, B, and C adequately represent the full diversity of 2D geometries. Moreover, all experiments are limited to 2D cases, leaving the scalability and applicability to 3D geometries unverified. At least one experiment on a 3D domain would have strengthened the validation of the proposed framework.
>
> Domains A, B and C represent three different topologies: complex shape with no hole, simple shape with two simple holes, simple shape with one complex hole. One can also apply our method to other more complicated shapes. We have done one more experiment on the dolphin shape domain to show the ability of our method in solving problems on complicated domains.
>
> Regarding 3D geometries, we  have one temporal 2D example (Heat 2D) which can be a special case of 3D problem. For general 3D problem where the spatial dimension is 3D, we list this as a major future work in the Conclusion and Future Works. The proposed iterative algorithm SNI can be extended without modification. However, implementation presents additional challenges in 3D, for example, generating building blocks in 3D. One way is to use polyhedrons as building blocks in 3D. Another choice is to generate building blocks from "growing" tetrahedra in 3D. Both can be very interesting ways to explore.
>
>
> > Since the computational cost heavily depends on the number of iterations required by the iterative inference process, efficiency can become a concern. Moreover, the number of subdomains Kmay depend on the number of vertices and edges, especially when field variations within the domain are significant. However, the authors treat Kas independent of the number of vertices and edges when estimating the time complexity, which can be misleading. This assumption suggests a linear relationship with the number of vertices, while in practice, the computational complexity may scale multiplicatively with both the number of vertices and the number of iterations.
>
> We thank the reviewer for raising this important point regarding the theoretical time complexity. The reviewer is correct that in a strict computational analysis, treating K as independent of v is a simplification, and a perfectly scalable implementation would indeed aim for K = O(v) to keep subdomain sizes constant, leading to a complexity of O(v N). We will revise the section 3.3 in the manuscript to include a discussion on this nuance, acknowledging the theoretical scaling while presenting the practical evidence that supports the robustness and utility of our method.
>
> However, we would like to clarify the practical context of our method, which mitigates the concern raised:
>
> 1. The Role of the Neural Operator as an Approximate Solver: The core of our method is the use of a neural operator as a fast, approximate local solver. Unlike traditional numerical solvers where the cost per subdomain (b) scales with subdomain size, a key advantage of a pre-trained neural operator is that its inference cost is largely determined by its architecture and is relatively stable for a range of subdomain sizes and complexities. This means that in practice, the relationship between K and v is more flexible and not strictly linear.
>
> 2. Empirical Evidence from Ablation Studies: Our primary concern is the final accuracy and convergence of the global solution, not just raw computational cost. As shown in our ablation studies (Figure 4), the final solution accuracy is remarkably robust to the choice of K. Once a sufficient minimum number of subdomains is reached to adequately represent the global geometry, increasing K further has a diminishing effect on the final result. This empirical finding is crucial: it means that in practice, K can be chosen based on geometric and accuracy considerations rather than being rigidly tied to v, preventing the multiplicative scaling from becoming prohibitive.
>
> In summary, while the reviewer's theoretical observation is valid for a classical solver, the use of a neural operator decouples the cost to some extent and introduces a different set of practical trade-offs. Our empirical results demonstrate that a good solution can be achieved without aggressively scaling K with v, thus the theoretical worst-case complexity does not fully capture the practical efficiency of our framework.

---

> ### Author Response · Authors · 2025-11-21
>
> > Although the proposed framework represents the computational domain as a graph, the paper does not include any comparison or discussion with graph-based neural networks or their variants (e.g., [1, 2, 3]), which could provide relevant baselines or complementary perspectives.
>
> The reviewer is right in that the GNN can provide additional perspective to our iterative SNI. We touched a bit on the graph-based neural networks in Appendix H. Our iterative algorithm SNI have many similarity with the message passing mechanism in GNN and there can be quite a lot to explore in improving the efficiency of the iterative scheme using GNN. This will be a very interesting future work.
>
> As comparison, we include one baseline with the implementation of MeshGraphNets from Learning Mesh-Based Simulation with Graph Networks, ICLR, 2021. We train the MeshGraphNets with our Laplace2D training data and perform direct inference. Please see the result in Appendix I.4. We do see that the GNN provides better generalization compared to GNOT and can be used to accelerate our iterative algorithm as future work.

---

### Author Response · Authors · 2025-12-02
**Summary for AC**

We summarize below the main concerns from each reviewer and how we addressed them during the rebuttal. Across all reviewers, we provided additional experiments, theoretical clarification, runtime analysis, and expanded related work.

## ***Reviewer HhMK***

**Main concerns:** geometry diversity and 3D scalability; complexity analysis; lack of GNN comparison.

**Our response:** We clarified that the three benchmark domains represent distinct topologies and added a more complex dolphin-shaped geometry demonstrating scalability. We refined Section 3.3 to explain that practical inference cost depends mainly on neural-operator evaluation, which remains stable across subdomain sizes; empirically, performance is robust to K. We also added a MeshGraphNets baseline and explained how SNI corresponds to a domain-decomposition form of message passing, showing GNNs can complement and accelerate the iterative scheme.

**Discussion:** We did not receive any feedback about our response before the rating was reverted.


## ***Reviewer 1tUX***

**Main concerns:** missing examples of simple training shapes; unclear preprocessing; high iteration count; limited impact of augmentation; BC encoding; 3D extension.

**Our response:** We added representative training-shape illustrations (Appendix I.3) and clarified that SNI uses a standard, fully automated pipeline (triangulation → METIS → overlap creation). We explained that iteration cost stems from unoptimized Python loops, while each neural-operator inference is extremely fast; detailed measurements are in Appendix G. We clarified that augmentation is not the central contribution and can have mixed effects. We explained differences in BC encoding between GNOT and GeoFNO. Finally, we outlined how SNI directly extends to 3D and discussed practical considerations for 3D building-block generation.

**Discussion:** We did not receive any feedback about our response before the rating was reverted.


# ***Reviewer bCzG (raised score to 6 after rebuttal)***

**Main concerns:** applicability (parabolic PDE wording), missing references, dataset availability, hyperparameters & convergence, scaling, error localization, baseline failure modes.

**Our response:** We clarified that the “parabolic” phrasing was misleading—the method is not limited to parabolic PDEs; SNI’s theory applies to general second-order, self-adjoint, coercive elliptic operators (e.g., Laplace, Poisson, Darcy). We added all missing references and explained why public datasets do not support geometry generalization; we will release our dataset and FEniCS-based pipeline. We elaborated on convergence conditions (operator assumptions + finite overlap) and clarified how K affects subdomain validity and iteration speed. We added results on a simple disk, provided detailed error visualizations, and analyzed cases where GNOT collapses to trivial solutions—showing that SNI corrects these via iterative coupling.

**Discussion:** Reviewer bCzG explicitly appreciated these clarifications and increased the score to 6.

# ***Reviewer kNbL***

**Main concerns:** runtime vs. classical solvers; complex-geometry testing; related work; generalization of the method.

**Our response:** We provided explicit timing (FEM  vs. one GNOT inference) and clarified that SNI serves as a DDM solver, beneficial for large/complex domains rather than small ones. We added results on a challenging dolphin-shaped geometry (40 subdomains) highlighting scalability. We strengthened related-work coverage (xPINN, cPINN, SNAP-DDM) and clarified that the framework is architecture- and physics-agnostic, with natural extension to more PDE types and mixed boundary conditions.

**Discussion:** We did not receive any feedback about our response before the rating was reverted.

---

### Meta-Review · Area_Chair_sG2A · 2026-01-06

**Summary:**

* The local-to-global framework enables neural operators to generalize to complex and unseen geometries by training only on simple basic shapes.
* The method demonstrates robust performance on domains with different topologies and holes where monolithic models fail.
* The Schwarz Neural Inference (SNI) algorithm provides a theoretically grounded iterative scheme with guaranteed convergence for elliptic PDEs.
* Training on local sub-problems significantly reduces the required number of data samples compared to direct global learning.
* The framework is architecture-agnostic and successfully integrates with multiple neural operator backbones like GNOT and Geo-FNO.
* The current implementation involves high inference overhead due to the large number of iterations required for convergence.

This paper proposes a framework that decomposes complex physical domains into simpler subdomains. A neural operator is trained on these basic shapes and then used within an iterative "Schwarz inference scheme" to reconstruct global solutions. This approach bypasses the memory and generalization limits of global models.

The submission is supported by both theoretical convergence analysis + empirical validation on various 2D PDEs. While the iterative nature of the solver introduces computational overhead, the ability to handle arbitrary geometries without retraining represents a step forward for the field. The authors provided additional experiments during the rebuttal that confirmed the method's scalability to more complex shapes.

**Reviewer Concerns:**

* [Core] Lack of diversity in test geometries: Affects the claim of arbitrary generalization; resolved by the addition of the complex dolphin-shaped geometry. (Reviewers HhMK, kNbL)
* [Core] Clarification of PDE scope: Affects the theoretical applicability; resolved by clarifying that the method is intended for elliptic and parabolic equations rather than being strictly limited. (Reviewer bCzG)
* [Non-core] Visualization of training shapes: Affects clarity of the data generation process; resolved by adding examples of simple polygons in the appendix. (Reviewer 1tUX)
* [Core] Comparison with GNNs: Affects baseline strength; resolved by adding a MeshGraphNets baseline showing the framework's comparative advantage. (Reviewer HhMK)
* [Non-core] Preprocessing steps: Affects reproducibility; resolved by detailing the use of METIS for automated domain partitioning. (Reviewer 1tUX)

### Still outstanding
* [Core] Inference time overhead: Affects practical utility; partially resolved by discussion, but the high iteration count remains a bottleneck compared to direct inference. (Reviewers 1tUX, kNbL)
* [Core] Extension to 3D domains: Affects claims of general scalability; not resolved as 3D implementation is deferred to future work. (Reviewer HhMK)

The rebuttal period was useful. Authors addressed major concerns regarding geometry diversity + the theoretical scope of the PDEs. They also provided important baseline comparisons that were initially missing. The primary remaining concern is the computational cost of the iterative process: the authors acknowledge as an area for code optimization rather than a fundamental flaw.

**Reviewer Scores:**

* **Reviewer HhMK**
* Original score: 4
* Estimated score shift: increase
* The reviewer's concerns about geometry diversity + GNN baselines were directly addressed with new experiments and data.

* **Reviewer 1tUX**
* Original score: 8
* Estimated score shift: unchanged
* This reviewer already liked the paper and recognized the strength of the stability and generalization results and was satisfied with the added visual evidence. Unlikely to lower the score.

* **Reviewer bCzG**
* Original score: 6
* Estimated score shift: unchanged (after increasing from 4)
* The reviewer engaged deeply with the technical failure modes of the baselines + reviewer was satisfied by the theoretical clarifications.

* **Reviewer kNbL**
* Original score: 4
* Estimated score shift: increase
* While the reviewer noted speed concerns, the authors' clarification on the memory benefits for large-scale problems provides a strong counter-argument that justifies the approach.

The reviewers initially expressed reservations about the narrow scope of the 2D benchmarks and the speed of the iterative solver. However, the rebuttal demonstrated that the framework solves cases where existing models collapse.

---

### Decision · Program_Chairs · 2026-01-26

Accept (Poster)